# Scaling Safe Learning-based Control to Long-Horizon Temporal Tasks

## Abstract

This paper introduces a model-based approach for training parameterized policies for an autonomous agent operating in a highly nonlinear (albeit deterministic) environment. We desire the trained policy to ensure that the agent satisfies specific task objectives and safety constraints, both expressed in Signal Temporal Logic. We assert that this learning problem is similar to training recurrent neural networks (RNNs), where the number of recurrent units is proportional to the temporal horizon of the agent's task objectives. This poses a challenge: RNNs are susceptible to vanishing and exploding gradients, and naïve gradient descent-based strategies to solve long-horizon task objectives thus suffer from the same problems. To tackle this challenge, we introduce a novel gradient approximation algorithm based on the idea of gradient sampling, and a smooth computation graph that provides a neurosymblic encoding of STL formulas. We show that these two methods combined improve the quality of the stochastic gradient, enabling scalable backpropagation over long time horizon trajectories. We demonstrate the efficacy of our approach on various motion planning applications requiring complex spatio-temporal and sequential tasks ranging over thousands of time steps.

## 1 Introduction

Learning-based approaches to synthesize control policies for highly nonlinear dynamical systems are prevalent across diverse domains, from autonomous vehicles to robots. Popular ways to train NN-based controllers include deep reinforcement learning (RL)(Berducci et al., 2021; Li et al., 2017; Chua et al., 2018; Srinivasan et al., 2020; Velasquez et al., 2021) and deep imitation learning (Fang et al., 2019). Techniques to synthesize neural controllers (including deep RL methods) largely focus on optimizing user-defined rewards or costs, but do not directly address specific spatio-temporal task objectives. For example, consider the objective that the system must reach region $R_1$ before reaching region $R_2$, while avoiding an obstacle region. Such spatio-temporal task objectives can be expressed in the formalism of Signal Temporal Logic (STL) (Maler & Nickovic, 2004). Furthermore, for any STL specification and a system trajectory, we can efficiently compute the *robustness degree*, or the approximate signed distance of the trajectory from the set of trajectories satisfying/violating the specification (Donzé & Maler, 2010; Fainekos et al., 2009).

The use of STL-based objectives has seen considerable recent interest in data-driven methods for training controllers for dynamical systems that can be described by (stochastic) difference equations. This literature brings together two separate threads: (1) smooth approximations to the robustness degree of STL specifications (Gilpin et al., 2020; Pant et al., 2017) enabling the use of STL robustness in gradient-based learning of control policies, and (2) efficient representation of the robustness computation allowing its use in training neural controllers using backpropagation (Yaghoubi & Fainekos, 2019; Leung et al., 2019; 2021; Hashemi et al., 2023; Hashemi et al.). We are inspired by the work in (Hashemi et al., 2023) that proposes a ReLU-based neural network encoding (called STL2NN) to exactly encode the STL robustness degree computation. We show how we can extend this computation graph to obtain smooth underapproximations of the STL robustness degree. Backpropagation-based methods typically treat the one-step environment dynamics and the neural controller as a recurrent unit that is then unrolled as many times as required by the temporal horizon of the specification $\varphi$. For instance, if enforcing $\varphi$ requires reasoning over several hundred time-steps, then it involves training a recurrent structure that resembles RNN with hundreds of recurrent units. It is well-known that training of RNNs over long sequences faces problems of exploding and vanishing

gradients (Goodfellow et al., 2016; Ba et al., 2016). To address this, we propose a sampling-based approximation of the gradient of the objective function (i.e. the STL property), that is particularly effective when dealing with behaviors over large time-horizons. Our method can improve training of NN controllers by at least an *order of magnitude*, i.e., in some cases, we reduce training times from hours to minutes. Several planning problems require finding optimal paths over long time-horizons. For example, consider the problem of planning the trajectory of a UAV in a complex, GPS-denied urban environment; here, it is essential that the planned trajectory span several minutes while avoiding obstacles and reaching several sequential goals (Windhorst et al., 2021).

**Contributions**. To summarize, we make the following contributions:

1. We propose smooth versions of computation graphs representing the robustness degree computation of an STL specification over the trajectory of a dynamical system. Our computation graph guarantees that it lower bounds the robustness degree with a tunable degree of approximation.
2. We develop a backpropagation framework which leverages the new differentiable structure, and we show how we can handle STL specifications.
3. We develop a sampling-based approach to approximate the gradient of STL robustness w.r.t. the NN controller parameters. Emphasizing the time steps that contribute the most to the gradient, our method randomly samples time points over the trajectory. We utilize the structure of the STL formula and the current system trajectory to decide which time-points represent critical information for the gradient.
4. We demonstrate the efficacy of our approach on high dimensional nonlinear dynamical systems involving long-horizon and dynamic temporal specifications.

**Related Work**. The use of temporal logic specifications for controller synthesis is a well-studied problem. Early work focuses on the model-based setting, where the environment dynamics are described either as Markov decision processes (Sadigh & Kapoor, 2016; Haesaert et al., 2018) or as differential equations (Gilpin et al., 2020; Pant et al., 2018; Raman et al., 2014; Farahani et al., 2015; Lindemann & Dimarogonas, 2018; Raman et al., 2015; Kalagarla et al., 2020; Lacerda et al., 2015; Guo & Zavlanos, 2018)). Recent years have also seen growing interest in data-driven techniques (Balakrishnan et al., 2022; Li et al., 2018) for control synthesis. In addition, automata-based approaches (Sadigh et al., 2014; Hasanbeig et al., 2018; Hahn et al., 2020; Lavaei et al., 2020) are also proposed in the field to address temporal logic based objectives. In (Liu et al., 2021), the authors propose an imitation learning framework where a Model-Predictive Controller (MPC) guaranteed to satisfy an STL specification is used as a teacher to train a recurrent neural network (RNN). In (Wang et al., 2023; Balakrishnan & Deshmukh, 2019), the authors replace handcrafted reward functions with the STL robustness within single-agent or multi-agent deep RL frameworks. The overall approach of this paper is the closest to the work in (Yaghoubi & Fainekos, 2019; Leung et al., 2019; 2021; Hashemi et al., 2023; Hashemi et al.), where STL robustness is used in conjunction with backpropagation to train controllers. The work in this paper makes significant strides in extending previous approaches to handle very long horizon temporal tasks, crucially enabled by the novel sampling-based gradient approximations. Due to the structure of our NN-controlled system, we can seamlessly handle time-varying dynamics and complex temporal dependencies.

The rest of the paper is organized as follows. In Sec. 2, we introduce the notation and the problem definition. We propose our learning-based control synthesis algorithms in Sec. 3, present experimental evaluation in Sec. 4, and conclude in Sec. 5

## 2 PRELIMINARIES

We use bold letters to indicate vectors and vector-valued functions, and calligraphic letters to denote sets. We denote the set, $\{1, 2, \cdots, n\}$ with $[n]$. A feed forward neural network (NN) with $\ell$ hidden layers is denoted by the array $[n_0, n_1, \cdots n_{\ell+1}]$, where $n_0$ denotes the number of inputs, $n_{\ell+1}$ is the number of outputs and for all $i \in [\ell]$, $n_i$ denotes the width of $i^{th}$ hidden layer.

**Neural Network Controlled Dynamical Systems (NNCS)**. Let $\mathbf{s} \in \mathbb{R}^n$ and $\mathbf{a} \in \mathbb{R}^m$ denote the state and action variables that take values from compact sets $\mathcal{S} \subseteq \mathbb{R}^n$ and $\mathcal{C} \subseteq \mathbb{R}^m$, respectively. We use $\mathbf{s}_k$ (resp. $\mathbf{a}_k$) to denote the value of the state (resp. action) at time $k$. We define a neural network

controlled system (NNCS) as a recurrent difference equation.

$$\mathbf{s}_{k+1} = \mathbf{f}(\mathbf{s}_k, \mathbf{a}_k). \tag{1}$$

We assume that the control policy is a parameterized function $\pi_\theta$, where $\theta$ is a vector of parameters that takes values in $\Theta$. Later in the paper, we instantiate the specific parametric form using a neural network for the controller. Given a fixed vector of parameters $\theta$, the parametric control policy $\pi_\theta$ returns an action $\mathbf{a}_k$ as a function of the current state $\mathbf{s}_k \in \mathcal{S}$ and time $k \in \mathbb{Z}^{\geq 0}$, or $\mathbf{a}_k = \pi_\theta(\mathbf{s}_k, k)$.

**Closed-loop Model Trajectory**. For a discrete-time NNCS as shown in equation 1, and a set of designated initial states $\mathcal{I} \subseteq \mathcal{S}$, under a pre-defined feedback policy $\pi_\theta$, equation 1 represents an autonomous discrete-time dynamical system. For a given initial state $\mathbf{s}_0 \in \mathcal{I}$, a system trajectory $\sigma_{\mathbf{s}_0}^\theta$ is a function mapping time instants in $[0, K]$ to $\mathcal{S}$, where $\sigma_{\mathbf{s}_0}^\theta(0) = \mathbf{s}_0$, and for all $k \in [0, K-1]$, $\sigma_{\mathbf{s}_0}^\theta(k+1) = \mathbf{f}(\mathbf{s}_k, \pi_\theta(\mathbf{s}_k, k))^1$. The computation graph for this trajectory is a recurrent structure. Appendix K shows an illustration of this structure and its similarity to RNN. In this paper, we provide algorithms to learn a policy $\pi_{\theta^\star}$ that maximizes the degree to which certain task objectives and safety constraints are satisfied. To that end, we formulate policy learning as an optimization problem.

**Task Objectives and Safety Constraints**. We assume that task objectives or safety constraints of the system are specified in a temporal logic known as Signal Temporal Logic (STL)(Maler & Nickovic, 2004). Our STL formulas are defined using the following syntax:

$$\varphi = h(\mathbf{s}) \bowtie 0 \mid \varphi_1 \wedge \varphi_2 \mid \varphi_1 \vee \varphi_2 \mid \mathbf{F}_I \varphi \mid \mathbf{G}_I \varphi \mid \varphi_1 \mathbf{U}_I \varphi_2 \tag{2}$$

that are limited to positive normal form logical expressions. Here, $\bowtie \in \{\leq, <, >, \geq\}$, $h$ is a function from $\mathcal{S}$ to $\mathbb{R}$, and $I$ is a closed interval $[a, b] \subseteq [0, K]$. The formal semantics of STL over discrete-time trajectories have been previously discussed in (Fainekos & Pappas, 2006), we briefly recall them here.

*Boolean Semantics and Formula Horizon*. We denote the formula $\varphi$ being true at time $k$ in trajectory $\sigma_{\mathbf{s}_0}^\theta$ by $\sigma_{\mathbf{s}_0}^\theta, k \models \varphi$. We say that $\sigma_{\mathbf{s}_0}^\theta, k \models h(\mathbf{s}) \bowtie 0$ iff $h(\sigma_{\mathbf{s}_0}^\theta(k)) \bowtie 0$. The semantics of the Boolean operations ($\wedge, \vee$) follow standard logical semantics of conjunctions and disjunctions, respectively. For temporal operators, we say $\sigma_{\mathbf{s}_0}^\theta, k \models \mathbf{F}_I \varphi$ is true if there is a time $k'$ that $k'-k \in I$ where $\varphi$ is true. Similarly, $\sigma_{\mathbf{s}_0}^\theta, k \models \mathbf{G}_I \varphi$ is true iff $\varphi$ is true for all $k'$ where $k'-k \in I$. In addition, $\sigma_{\mathbf{s}_0}^\theta, k \models \varphi_1 \mathbf{U}_I \varphi_2$ if there is a time, $k', k'-k \in I$ where $\varphi_2$ is true and for all times $k'' \in [k, k')$ $\varphi_1$ is true. The temporal scope or horizon of an STL formula defines the number of time-steps required in a trajectory to evaluate the formula, $\sigma_{\mathbf{s}_0}^\theta, 0 \models \varphi$ (Maler & Nickovic, 2004). For example, the temporal scope of the formula $\mathbf{F}_{[0,3]}(x > 0)$ is 3, and that of the formula $\mathbf{F}_{[0,3]}\mathbf{G}_{[0,9]}(x > 0)$ is $3 + 9 = 12$.

*Quantitative Semantics (Robustness value) of STL*. Quantitative semantics of STL roughly define a signed distance of a given trajectory from the set of trajectories satisfying or violating the given STL formula. There are many alternative semantics proposed in the literature (Donzé & Maler, 2010; Fainekos & Pappas, 2006; Rodionova et al., 2022; Akazaki & Hasuo, 2015); in this paper, we focus on the semantics from (Donzé & Maler, 2010) that are shown below. The robustness value $\rho(\varphi, \sigma_{\mathbf{s}_0}^\theta, k)$ of an STL formula $\varphi$ over a trajectory $\sigma_{\mathbf{s}_0}^\theta$ at time $k$ is defined recursively as follows[2].

$$
\begin{array}{|ll|ll|}
\hline
\varphi & \rho(\varphi, k) & \varphi & \rho(\varphi, k) \\
\hline
h(\mathbf{s}_k) \geq 0 & h(\mathbf{s}_k) & \mathbf{F}_{[a,b]}\psi & \max_{k' \in [k+a, k+b]} \rho(\psi, k) \\
\varphi_1 \wedge \varphi_2 & \min(\rho(\varphi_1, k), \rho(\varphi_2, k)) & & \\
\varphi_1 \vee \varphi_2 & \max(\rho(\varphi_1, k), \rho(\varphi_2, k)) & \varphi_1 \mathbf{U}_{[a,b]}\varphi_2 & \max_{k' \in [k+a, k+b]} \left( \min \left( \begin{array}{l} \rho(\varphi_2, k'), \\ \min_{k'' \in [k,k')} \rho(\varphi_1, k'') \end{array} \right) \right) \\
\mathbf{G}_{[a,b]}\psi & \min_{k' \in [k+a, k+b]} \rho(\psi, k) & & \\
\hline
\end{array}
\tag{3}
$$

We note that if $\rho(\varphi, k) > 0$ the STL formula $\varphi$ is satisfied at time $k$, and we say that the formula $\varphi$ is satisfied by a trajectory if $\rho(\varphi, 0) > 0$.

*STL Robustness as a* ReLU *NN*. The quantitative semantics in equation 3 contains $\min/\max$ operators; this makes the robustness of an STL formula difficult to be used in gradient-based methods for learning.

---

[1]If the policy $\pi_\theta$ is obvious from the context, we drop the $\theta$ in the notation $\sigma_{\mathbf{s}_0}^\theta$.

[2]For brevity, we omit the trajectory from the notation, as it is obvious from the context.

However, $\min/\max$ operators in equation 3 can be expressed using ReLU functions as follows:

$$\min(a_1, a_2) = a_1 - \mathsf{ReLU}(a_1 - a_2), \quad \max(a_1, a_2) = a_2 + \mathsf{ReLU}(a_1 - a_2). \tag{4}$$

This allows the computation graph representing the robustness of an STL formula w.r.t. a given trajectory to be expressed using repeated application of the ReLU function (with due diligence in balancing $\min, \max$ computations over several arguments into a tree of at most logarithmic height in the number of operands). We call this ReLU-based computation graph as STL2NN. The STL2NN, despite being reformulated with ReLU, is essentially equivalent to non-smooth robustness in equation 3, making it unsuitable for back-propagation. To address this, smooth activations are introduced to create a differentiable computation graph.

## 3 TRAINING NEURAL NETWORK CONTROL POLICIES

**Problem Definition**.: We wish to learn a neural network (NN) control policy $\pi_\theta$ (or equivalently the parameter values $\theta$), s.t. for any initial state $\mathbf{s}_0 \in \mathcal{I}$ [3], using the control policy $\pi_\theta$, the trajectory obtained, i.e., $\sigma_{\mathbf{s}_0}^\theta$ satisfies a given STL formula $\varphi$.

Our solution strategy is to treat each time-step of the given dynamical equation in equation 1 as a recurrent unit. We then sequentially compose or unroll as many units as required by the horizon of the STL specification. For instance, if the specification is $\mathbf{F}_{[0,10]}(x > 0)$, then, we use 10 instances of $f(\mathbf{s}_k, \pi_\theta(\mathbf{s}_k))$ by setting the output of the $k^{th}$ unit to be the input of the $(k+1)^{th}$ unit. This unrolled structure implicitly contains the system trajectory, $\sigma_{\mathbf{s}_0}^\theta$ starting from some initial state $\mathbf{s}_0$ of the system. The unrolled structure essentially represents the symbolic trajectory, where each recurrent unit shares the NN parameters of the controller (see Appendix K for more detail). By composing this structure with the neural network representing the given STL specification $\varphi$; for instance, the STL2NN computation graph introduced in the previous section, we have a NN that maps the initial state of the system in equation 1 to the robustness degree of $\varphi$. Thus, training the parameters of this resulting NN to guarantee that its output is positive (for all initial states) guarantees that each system trajectory satisfies $\varphi$. However, we face two main challenges in training such a NN.

*Challenge 1*: The cost function to be optimized is the output of the STL2NN computation graph. As mentioned earlier, as this is identical to the non-smooth robustness proposed in equation 3, we cannot use it effectively with stochastic optimization frameworks. An obvious step is to approximate STL2NN by a smooth function. We represent this function as STL2LB and leverage it for computing the gradients of the robustness function. It is important for STL2LB to lower bound STL2NN; if we find NN parameters that guarantee a positive output of STL2LB for all possible system trajectories, then it guarantees that the system satisfies the given STL objective.

*Challenge 2:* As our model can be thought of as a recurrent structure with number of repeated units proportional to the horizon of the formula, naïve gradient-based training algorithms are applicable to only short time horizons. As our structure is recurrent, the gradient computation faces the same issues of vanishing and exploding gradients when dealing with long trajectories that RNNs may face in training (Pascanu et al., 2013). We introduce an efficient technique to approximate gradients for long trajectories that is inspired by the idea of Drop-out (Srivastava et al., 2014). This popular technique also suggests us calling this approximate gradient as ***robust gradient***.

### 3.1 SMOOTH, GUARANTEED LOWER BOUND FOR STL2NN

To guarantee a smooth lower bound for STL2NN, we replace ReLU activations in the $\min$ operation with the softplus activation function defined as:

$$\mathsf{softplus}(a_1 - a_2) = \frac{1}{b} \log\left(1 + e^{b(a_1 - a_2)}\right), \; b > 0.$$

Similarly we replace the ReLU activation functions contributing in $\max$ operation with the swish activation function:

$$\mathsf{swish}(a_1 - a_2) = \frac{a_1 - a_2}{1 + e^{-b(a_1 - a_2)}}, \; b > 0.$$

---

[3]In the context of neural network training we satisfy this condition considering a set of sampled initial states, but we verify our trained NN for all the initial states through formal verification techniques.

We denote this smooth NN with STL2LB and we claim: (see Appendix J for more detail)

$$\forall (\sigma_{\mathbf{s}_0}, b) \in \mathbb{R}^{nK} \times \mathbb{R} \ : \ \mathsf{STL2LB}(\sigma_{\mathbf{s}_0}; b) \leq \mathsf{STL2NN}(\sigma_{\mathbf{s}_0})$$

We note that replacing the $\min$ and $\max$ operators with smooth versions is, by itself, *not* novel. Several prior studies have explored smooth semantics for STL (Gilpin et al., 2020; Pant et al., 2017). For example, consider the smooth $\max$ and $\min$ introduced in (Gilpin et al., 2020; Pant et al., 2017; Liu et al., 2021; Leung et al., 2019; Lindemann & Dimarogonas, 2018):

$$\widetilde{\max}(a_1, \cdots, a_\ell) = \underbrace{\frac{1}{b} \log \left( \sum_{i=1}^{\ell} e^{ba_i} \right)}_{\text{Logexpsum}} \quad \text{or} \quad \widetilde{\max}(a_1, \cdots, a_\ell) = \underbrace{\sum_{i=1}^{\ell} \frac{a_i e^{ba_i}}{\sum_{i=1}^{\ell} e^{ba_i}}}_{\text{Boltzmann}}. \tag{5}$$

and $\widetilde{\min}(a_1, \cdots, a_\ell) = -\widetilde{\max}(-a_1, \cdots, -a_\ell)$.

An issue with using any kind of smooth approximation is that numerical issues can be caused by the presence of large positive exponents. Here, we explain this with an example.

**Example 1.** Let $a_1 = 0$, and $a_2 = 80$, and suppose we wish to perform a smooth approximation of $\max(a_1, a_2)$ with Logexpsum, Boltzmann and swish operators. Let the parameter $b = 10$. Then we can see that computing $\exp(ba_2)$ and $\exp(-b(a_1 - a_2))$ causes numerical issues. On the other hand, for $a_1 = 80, a_2 = 0$ the softplus operator may also fail. $\square$

Hence, to resolve the computation problem, we can define a threshold $\tau > 0$ large enough and approximate swish and softplus activation functions as:

$$\widetilde{\mathsf{swish}}(\zeta) = \left\{ \begin{array}{ll} \mathsf{swish}(\zeta) & \text{if } \zeta > -\tau/b \\ 0 & \text{if } \zeta < -\tau/b \end{array} \right., \qquad \widetilde{\mathsf{softplus}}(\zeta) = \left\{ \begin{array}{ll} \zeta & \text{if } \zeta > \tau/b \\ \mathsf{softplus}(\zeta) & \text{if } \zeta < \tau/b, \end{array} \right.$$

where $\zeta = a_1 - a_2$. It is important to note that such a technique cannot be performed for smoothing using Logexpsum or Boltzmann-style operators and is exclusively applicable on STL2LB. By selecting $\tau$ large enough, we can maintain the differentiability of operators, at least to the accuracy level of existing computation tools. To avoid the shortcomings of Logexpsum and Boltzmann-style approximations, we use softplus (with the above modifications) and the swish function as activations.

**Lemma 1.** *For any formula $\varphi$ belonging to STL in positive normal form, and $b > 0$, for a given trajectory $\sigma_{\mathbf{s}_0} = \mathbf{s}_0, \mathbf{s}_1, \ldots, \mathbf{s}_K$, if $\mathsf{STL2LB}(\sigma_{\mathbf{s}_0}; b) > 0$, then $\sigma_{\mathbf{s}_0} \models \varphi$, where $\mathsf{STL2LB}$ is a computation graph for STL robustness degree but with the modified softplus activation instead of $\min$ and the modified swish activation instead of $\max$.*

See Appendix J for proof. The main contributions of STL2LB comparing to the existing smooth robustness formula (Gilpin et al., 2020; Pant et al., 2017) can be summarized as follows:

- Example 1 shows that STL2LB provides convenience for computation.
- Lemma 1 indicates that, like (Gilpin et al., 2020), it is also a guaranteed smooth lower-bound for robustness function, thus, can be considered as a control barrier function.

## 3.2 TRAINING WITH STL2LB

In order to train the controller for all initial states, $\mathbf{s}_0 \in \mathcal{I}$ we solve the following optimization problem:

$$\theta^* = \arg\max_{\theta} \left( \mathbb{E}_{\mathbf{s}_0 \overset{u}{\sim} \mathcal{I}} \left[ \rho(\varphi, \sigma_{\mathbf{s}_0}^{\theta}, 0) \right] \right),$$

$$\text{s.t. } \sigma_{\mathbf{s}_0}^{\theta}(k+1) = \mathbf{f}\left( \sigma_{\mathbf{s}_0}^{\theta}(k), \pi_{\theta}\left( \sigma_{\mathbf{s}_0}^{\theta}(k), k \right) \right).$$

that aims to increase the expectation of the robustness for initial states uniformly sampled from the set of initial states. Solving

---

**Algorithm 1:** Neurosymbolic policy learning

1 **Input:** $\widehat{\mathcal{I}}$, $\theta^0$, $b$, $\varphi$, $\bar{\rho}$
2 $j \leftarrow 0$
3 **while** $\left( \min_{\mathbf{s}_0 \in \widehat{\mathcal{I}}} \left( \rho(\varphi, \sigma_{\mathbf{s}_0}^{\theta^j}, 0) \right) < \bar{\rho} \right)$ **do**
4      $\mathbf{s}_0 \leftarrow$ Sample from $\widehat{\mathcal{I}}$
5      $\sigma_{\mathbf{s}_0}^{\theta^j} \leftarrow$ Simulate using policy $\pi_{\theta^j}$
6      $d \leftarrow \nabla_{\theta} \mathsf{STL2LB}(\sigma_{\mathbf{s}_o}^{\theta^j})$ using $\sigma_{\mathbf{s}_o}^{\theta^j}$
7      $\theta^{j+1} \leftarrow \theta^j + \mathsf{Adam}(d)$
8      $j \leftarrow j + 1$

---

this optimization problem is equivalent to training the NN controller using a gradient-based algorithm (shown in Alg. 1). However we terminate the algorithm once the robustness is above a pre-specified lower threshold $\bar{\rho}$. We also generate a population of samples from the set of initial states of the system, i.e. $\mathcal{I}$, for training purposes and denote this set by $\widehat{\mathcal{I}}$.

### 3.3 Extension to Long Horizon Temporal Tasks & Higher Dimensional Systems

When dealing with long time-horizon trajectories or high dimensional models, considering the entire trajectory to compute $\nabla_\theta \text{STL2LB}(\sigma_{\mathbf{s}_0}^{\theta^j})$ in Alg. 1, becomes computationally impractical as it either approaches zero (vanishes) or diverges (explodes) due to the high number of steps in the trajectory $\sigma_{\mathbf{s}_0}$. To alleviate this, inspired by the well-known idea of Drop-out (Srivastava et al., 2014) for backpropagation, we propose a sampling-based gradient approximation technique that prevents the gradient to explode/vanish and is also known to provide a robust training process. The basic idea in sampling-based technique is to only select certain time-points in the trajectory for gradient computation, while using a fixed older control policy at the non-selected points. In order to select time points, a naïve strategy is to choose time-points randomly. However, in our preliminary results, exploiting the structure of the given STL formula – specifically identifying and using *critical predicates* – gives superior results compared to random sampling.

**Definition 1** (Critical Predicate). *As the robustness degree of STL is an expression consisting of* $\min$ *and* $\max$ *of robustness values of predicates at different times, the robustness degree is consistently equivalent to the robustness of one of the predicates $h(\cdot)$ at a specific time. This specific predicate $h^*$ is called the critical predicate, and this specific time $k^*$ is called the critical time.*

A difficulty in using critical predicates is that a change in controller parameter values may change the system trajectory, which may in turn change the predicate that is critical for its robustness value. Specifically, if the critical predicate in one gradient step is different from the critical predicate in the subsequent gradient step, our gradient ascent strategy fails to augment the robustness value, since it only results in the elevation of that specific critical predicate's value. The incorrect gradient generated in this gradient step can lead to failure in the training process, as it may abruptly reduce the robustness value drastically.

Given a predefined specification $\varphi$, Fig. 1 shows the non-differentiable points in robustness as a function of control parameters, with each smooth segment corresponding to a distinct critical predicate. In order to optimize robustness within these smooth partitions, stochastic optimizers like Adam can be employed effectively. However, it is essential to note that the Adam optimizer's applicability is confined to differentiable points. To overcome this challenge, we employ a technique which utilizes STL2LB to re-smooth the problem at the non-differentiable local maxima. However, it is practically impossible to accurately detect the non-differentiable local maxima, thus we take a more conservative approach and shift the training approach to utilize STL2LB at every gradient step where the critical predicate technique is unable to improve the robustness. The rest of this section presents a detailed explanation for each module in our training algorithm, and Alg. 2 encapsulates these modules within a unified training process. In this algorithm, we use $\rho^\varphi(\sigma_{\mathbf{s}_0}^\theta)$ as shorthand for the robustness degree of $\sigma_{\mathbf{s}_0}^\theta$ w.r.t. $\varphi$ at time 0. A detailed explanation for Alg. 2 is also provided in Appendix A.

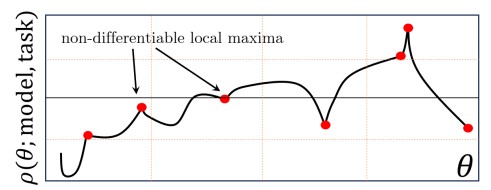

Figure 1: Shows a demonstration for the functionality of non-differentiable robustness function with respect to the control parameters. Assuming a fixed initial state, every control parameter is corresponding to a simulated trajectory, and that trajectory represents a robustness value. This robustness value is equal to the quantitative semantics for the critical predicate. In every single smooth part of this plot, the control parameters are offering a unique critical predicate.

**Sampling-based gradient approximation technique**. This technique is based on sampling across recurrent units and is originally inspired by the popular idea of Drop-out proposed in (Srivastava et al., 2014). Considering the NN controllers rolled out over the trajectory, the idea of Drop-out suggests removing the randomly selected nodes from a randomly selected NN controller over the trajectory. This requires the node to be absent in both forward-pass and backward-pass in backpropagation algorithm. However, our primary goal is to alleviate the problem of vanishing and exploding gradients. Thus, we propose to sample random time steps and select all of its controller nodes to apply Drop-out. However, for long trajectories we need to drop out a large portion of time steps that result in inaccurate approximation, thus we compensate for this by repeating this process and computing for accumulative gradients (See parameters $N_1, N_2$ in Alg. 2). Restriction of Drop-out to sample time steps results in less number of self multiplication of weights and therefore alleviates the problem of

vanishing/exploding gradient. However, this may result in disconnection between the trajectory states and thus we need to apply modifications to this strategy. To that end, we drop out the selected nodes but we also replace that group of selected nodes ( controller unit) with its evaluation in forward pass. This strategy motivates us to define the sampled trajectory as proposed in definition 2.

**Definition 2** (Sampled Trajectory). *Consider the set of time steps $\mathcal{T} = \{0, t_1, t_2, \cdots, t_N\}$ sampled from the horizon $\mathcal{K} = \{0, 1, 2, \cdots, K\}$, and the control parameters $\theta^j$ in the gradient step $j$. The sampled trajectory $\tilde{\sigma}^{\theta^j}_{\mathbf{s}_0, \mathcal{T}}$ is a subset of trajectory states $\sigma^{\theta^j}_{\mathbf{s}_0}$, where $\tilde{\sigma}^{\theta^j}_{\mathbf{s}_0, \mathcal{T}}(0) = \mathbf{s}_0$ and,*

$$\forall i \in \{0, 1, \cdots, N\}: \quad \tilde{\sigma}^{\theta^j}_{\mathbf{s}_0, \mathcal{T}}(i+1) = \mathbf{f}^j_i(\tilde{\sigma}^{\theta}_{\mathbf{s}_0, \mathcal{T}}(i), \pi_{\theta^j}(\tilde{\sigma}^{\theta^j}_{\mathbf{s}_0, \mathcal{T}}(i), t_i)).$$

*Given the pre-computed constants $\{\mathbf{a}_{1+t_i}, \mathbf{a}_{2+t_i}, \cdots \mathbf{a}_{t_{i+1}-1}\}$ using $\theta^j$ in the gradient step $j$, the dynamics model $\mathbf{f}^j_i$ is defined as: $\mathbf{f}^j_i(\mathbf{s}, \mathbf{a}) = \mathbf{f}(\mathbf{f}(\cdots(\mathbf{f}(\mathbf{s}, \mathbf{a}), \mathbf{a}_{1+t_i}), \mathbf{a}_{2+t_i}), \cdots, \mathbf{a}_{t_{i+1}-1}).$*

---

**Algorithm 2:** Gradient-direction approximation algorithm for training the controller for long horizon tasks.

1  **Input:** $\epsilon$, $M$, $N$, $N_1$, $N_2$, $\theta^0$, $\varphi$, $\bar{\rho}$, $\widehat{\mathcal{I}}$, $j = 0$

2  **while** $\rho^\varphi(\sigma^{\theta^j}_{\mathbf{s}_0}) \leq \bar{\rho}$ **do**

3  $\quad$ $\mathbf{s}_0 \leftarrow$ Sample from $\widehat{\mathcal{I}}$

4  $\quad$ use_STL2LB $\leftarrow$ False; $\quad j \leftarrow j + 1$

5  $\quad$ **if** use_STL2LB $=$ False **then**

6  $\quad\quad$ $\theta_1, \theta_2 \leftarrow \theta^j$

7  $\quad\quad$ **for** $i \leftarrow 1, \cdots, N_1$ **do**

8  $\quad\quad\quad$ $\sigma^{\theta^j}_{\mathbf{s}_0}, k^*, h^*(\mathbf{s}_{k^*}) \leftarrow$ Simulate trajectory, obtain critical predicate

9  $\quad\quad\quad$ $\mathcal{T}^q, X^q, \tilde{\sigma}^{\theta^j}_{\mathbf{s}_0, \mathcal{T}^q}, q \in [M] \leftarrow$ Generate sampled time steps & sampled trajectories

10  $\quad\quad\quad$ $d_1 \leftarrow$ robust gradient $\nabla_\theta \mathcal{J}^{wp}(\sigma^{\theta^j}_{\mathbf{s}_0})$

11  $\quad\quad\quad$ $d_2 \leftarrow$ robust gradient $\nabla_\theta h^*(\mathbf{s}_{k^*})$

12  $\quad\quad\quad$ $\theta_1 \leftarrow \theta_1 + \text{Adam}(d_1/N_1)$

13  $\quad\quad\quad$ $\theta_2 \leftarrow \theta_2 + \text{Adam}(d_2/N_1)$

14  $\quad\quad$ **if** $\rho^\varphi(\sigma^{\theta_1}_{\mathbf{s}_0}) \geq \rho^\varphi(\sigma^{\theta^j}_{\mathbf{s}_0})$ **then** $\theta^{j+1} \leftarrow \theta_1$

15  $\quad\quad$ **else if** $\rho^\varphi(\sigma^{\theta_2}_{\mathbf{s}_0}) \geq \rho^\varphi(\sigma^{\theta^j}_{\mathbf{s}_0})$ **then** $\theta^{j+1} \leftarrow \theta_2$

16  $\quad\quad$ **else**

17  $\quad\quad\quad$ $\ell \leftarrow 1, \quad$ update $\leftarrow$ True

18  $\quad\quad\quad$ **while** update & (use_STL2LB$=$False) **do**

19  $\quad\quad\quad\quad$ $\ell \leftarrow \ell/2; \hat{\theta} \leftarrow \theta^j + \ell(\theta_2 - \theta^j)$

20  $\quad\quad\quad\quad$ **if** $\rho(\varphi, \sigma^{\hat{\theta}}_{\mathbf{s}_0}, 0) \geq \rho^\varphi(\sigma^{\theta^j}_{\mathbf{s}_0})$ **then**

21  $\quad\quad\quad\quad\quad$ $\theta^{j+1} \leftarrow \hat{\theta}, \quad$ update $\leftarrow$ False

22  $\quad\quad\quad\quad$ **else if** $\ell < \epsilon$ **then** use_STL2LB $\leftarrow$ True

23  $\quad$ **if** use_STL2LB $=$ True **then**

24  $\quad\quad$ $\theta_3 \leftarrow \theta^j$

25  $\quad\quad$ **for** $i \leftarrow 1, \cdots, N_2$ **do**

26  $\quad\quad\quad$ $\mathcal{T}^q, X^q, \tilde{\sigma}^{\theta^j}_{\mathbf{s}_0, \mathcal{T}^q}, q \in [M] \leftarrow$ Generate sampled time steps & sampled trajectories

27  $\quad\quad\quad$ $d_3 \leftarrow$ robust gradient$\nabla_\theta \text{STL2LB}(\sigma^{\theta^j}_{\mathbf{s}_0}, b)$

28  $\quad\quad\quad$ $\theta_3 \leftarrow \theta_3 + \text{Adam}(d_3/N_2)$

29  $\quad\quad$ $\theta^{j+1} \leftarrow \theta_3$

---

Figure 2 in Appendix A makes this definition more clear through visualization. This definition applies the idea of Dropout that is also equipped with our modification to replace the set of selected nodes on a randomly selected time step with its pre-computed output in the forward pass for original trajectory. This set of nodes are indeed a controller unit on the sampled time step. However our contribution from the idea of sampled trajectory are listed as follows:

1. to apply the idea of Drop-out on control synthesis over extended trajectories which alleviates for the problem of vanishing/exploding gradients.

2. to restrict the sampling process to time-steps instead of a random node selection on trajectory.

3. to assure that the critical time is included in the set of sampled time steps.

In this work we denote the gradient of original trajectory with *'original gradient'* and the approximate gradient from our sampling technique as *'robust gradient'* [4]. In the backpropagation algorithm at a given gradient step $j$ with control parameter, $\theta^j$ we wish to compute the robust gradient $\partial \mathcal{J}/\partial \theta^j$. To that end, we utilize $\theta^j$ to simulate the trajectory $\{\mathbf{s}_0, \mathbf{s}_1, ..., \mathbf{s}_K\}$ and control sequence $\{\mathbf{a}_0, \mathbf{a}_1, ..., \mathbf{a}_{K-1}\}$. We then generate a set of random selections for the sampled times $\mathcal{T}^q, q \in [M]$ and define the sampled trajectories, $\tilde{\sigma}^{\theta^j}_{\mathbf{s}_0, \mathcal{T}^q}$ with the specified interrelation proposed in the definition 2. In the next gradient step, $j + 1$ we again generate a new set of sampled times and repeat the process. [5].

---

[4] We call this gradient robust since the Drop-out technique claims this gradient results in robust training.

[5] In this work, we evaluate the applicability of our sampling based technique through different case studies. This is a common approach to replace the mathematical proofs with validation through experimental results. See the famous works like (Srivastava et al., 2014).

**Way Point Function**. The way point function, $\mathcal{J}^{wp}(\sigma_{\mathbf{s}_0}^\theta)$, is established as a reward-based function designed to offer incentives to the optimizer to guide the trajectory toward a pre-defined path.

**Safe re-smoothing**. As discussed before, in the event that the optimization process steers the control parameters towards non-differentiable local maxima, there may be a drastic reduction in the value of the robustness function. In this case, we replace the objective function with $\mathcal{J}(\sigma_{\mathbf{s}_0}^{\theta^j}) = \mathsf{STL2LB}(\sigma_{\mathbf{s}_0}^{\theta^j}; b)$. This is because, $\mathsf{STL2LB}$ is a smooth version of robustness over the trajectory, in addition, it is a guaranteed lower bound for robustness and its distance to robustness can also be controlled with $b$. Thus, its inclusion makes the re-smoothing process safe against a potential drastic drop in robustness value.

In case the objective function $\mathcal{J}$ is the value of critical predicate, it is only a function of the trajectory state $\mathbf{s}_{k^*}$ and we sample the time steps as, $\mathcal{T} = \{0, t_1, t_2, \cdots, t_N\}$, $t_N = k^*$. The original gradient is $\partial\mathcal{J}/\partial\theta = (\partial\mathcal{J}/\partial\mathbf{s}_{k^*})(\partial\mathbf{s}_{k^*}/\partial\theta)$ but based on our sampling technique inspired with Drop-out, the robust gradient will be defined as, $\partial\mathcal{J}/\partial\theta = (\partial\mathcal{J}/\partial\mathbf{s}_{k^*})(\partial\tilde{\sigma}_{\mathbf{s}_0,\mathcal{T}}^\theta(N)/\partial\theta)$ where unlike $\partial\mathbf{s}_{k^*}/\partial\theta$ that is prone to vanish/explode problem, the new term $\partial\tilde{\sigma}_{\mathbf{s}_0,\mathcal{T}}^\theta(N)/\partial\theta$ can be computed efficiently[6].

In case the objective function is way-point or $\mathsf{STL2LB}$, that is a function of all the trajectory states, we consequently segment the trajectory into $M$ different partitions, by random time sampling as,

$$\mathcal{T}^q = \{0, t_1^q, t_2^q, \cdots, t_N^q\}, \; q \in [M], \; (\forall q_1, q_2 \in [M]: \mathcal{T}^{q_1} \cap \mathcal{T}^{q_2} = \{0\}) \wedge (\mathcal{K} = \bigcup_{q=1}^M \mathcal{T}^q), \quad (6)$$

with sub-trajectories generated by $\mathcal{T}^q, q \in [M]$ denoted as $X^q = \left\{\mathbf{s}_0, \mathbf{s}_{t_1^q}, \cdots, \mathbf{s}_{t_N^q}\right\}$. We know the original gradient in this case is $\partial\mathcal{J}/\partial\theta = \sum_{q=1}^M (\partial\mathcal{J}/\partial X^q)(\partial X^q/\partial\theta)$. However in our training process to compute the robust gradient, the gradient matrix $\partial X^q/\partial\theta$ is supposed to be replaced with $\partial\tilde{\sigma}_{\mathbf{s}_0,\mathcal{T}^q}^\theta/\partial\theta$. Unlike the inefficient gradient matrix $\partial X^q/\partial\theta$ that is prone to vanish/explode problem, the gradient matrix $\partial\tilde{\sigma}_{\mathbf{s}_0,\mathcal{T}^q}^\theta/\partial\theta$ can be computed efficiently.

## 4 EXPERIMENTAL EVALUATION

In this section, we evaluate the performance of our proposed method. We implemented all experiments in MATLAB[7]. We give the details of our experimental setup in the Appendix. We evaluate on 5 environments (details given in the Appendix) (a) a 3 dimensional simple car, (b) a 6 dimensional drone, (c) a 6 dimensional drone combined with a moving frame with a task requiring a long path plan, (d) a multi-agent system of 10 connected Dubins car, and (e) a 12 dimensional quad-rotor.

**Evaluation metric**. To evaluate the performance of our method, we first compare the results of Alg. 1 with the examples proposed in (Yaghoubi & Fainekos, 2019) for environments (a) and (b), and compare the runtimes. As the dimension of system increases, it becomes more challenging to avoid the training procedure from converging to local optima. Increasing the horizon of temporal task causes the gradients to become non-informative, as they potentially vanish or explode. Therefore, environments (c), (d) and (e) are solved with Alg. 2. We also show that Alg. 1 is unable to finish the computation for long horizon experiments within a reasonable number of iterations or runtime.

**Comparison**. Application of Alg. 1 on the environments (a) and (b), shows noticeable improvement, w.r.t. the previous work in (Yaghoubi & Fainekos, 2019). In these examples, we started from a random initial guess for NN parameters and computed the solution within $\approx 6$ minutes. However the reported runtime in (Yaghoubi & Fainekos, 2019) is noticeably higher than ours. Appendix L shows a comparison between the performance of $\mathsf{STL2LB}$ and the previous works (Pant et al., 2017; Gilpin et al., 2020). This comparison emphasizes on the computational problem proposed in Example 1.

**Main results**. We test the performance of our proposed sampling-based algorithm in highly nonlinear and high dimensional environments over long and also complex temporal tasks (details in the appendix). Table 2 reports the results of these experiments.

---

[6]The efficiency results from the control parameters $\theta$ repeating in fewer steps as most of them are fixed.

[7]All experiments were run on a laptop PC with a Core i9 CPU, and we did not utilize GPUs for computation.

To evaluate the contribution of Alg. 2 we perform an ablation study on a simple Dubin's car environment. We assume an $1m \times 1m$ area for execution, and specify that the car moves in this area within $K = 10$ time steps ($\delta t = 0.1$) while avoiding an obstacle presented in this area (Figure 11 is a scaled ($\times 100$) version of this area). We evaluate the same case study, but with task horizons ranging from 10 to 1000 time steps. With increasing number of time-steps, we also need to magnify the size of the environment to maintain task difficulty. The ablation study involves solving each of these problems: (1) with the vanilla version of Alg. 1 with no sampling-based robust gradient computation (2) Alg. 1 where sampling-based robust gradient approch is performed using random times within the trajectory, and (3) Alg. 2 that combines gradient-based sampling based on critical predicates, safe re-smoothing, and waypoint functions. We summarize the results in Table 1. We can see that the inclusion of time sampling decreases the runtime for training process. We also observe that for relatively small horizons $K = 10, 50$, Alg. 1 performs slightly better than Alg. 2 in terms of runtime but for $K = 100, 500, 1000$ Alg. 2 is much more efficient. In the table, an entry "NF" indicates when the algorithm is unable to solve the problem within 8000 gradient steps. In Alg. 1, as the dimension of STL2LB grows with the length of the horizon and dimension of the system, we see it struggle with the more complex case studies.

Table 2 highlights the versatility of our technique to handle various case studies with number of dimensions as high as 20, and time horizons in thousands of steps. We also use a diverse set of temporal task objectives that include nested temporal operators, and those involving trajectories from two independently moving objects (Drone & Moving Frame case study). The results were produced using Alg. 2.

| Horizon | Algorithm 1 (No time Sampling) | | Algorithm 1 (With time Sampling) | | Algorithm 2 (With time Sampling) | |
|---|---|---|---|---|---|---|
| | Num. of Iterations | Runtime (seconds) | Num. Iterations | Runtime (seconds) | Num. of Iterations | Runtime (seconds) |
| 10 | 34 | 2.39 | 11 | 1.39 | 4 | 5.61 |
| 50 | 73 | 2.46 | 53 | 14.01 | 25 | 6.09 |
| 100 | 152 | 8.65 | 105 | 112.6 | 157 | 90.55 |
| 500 | NF[$-1.59$] | 4986 | 3237 | 8566 | 624 | 890.24 |
| 1000 | NF[$-11.49$] | 8008 | NF[$-88.42$] | 28825 | 829 | 3728 |

Table 1: Ablation study. We mark the experiment with NF[.] if it is unable to provide a positive robustness within 8000 iterations, and the value inside brackets is the maximum value of robustness it finds. We magnify the environment proportional to the horizon (see Appendix H for details). All experiments use a unique guess for initial parameter values.

| Case Study | Temporal Task | System Dimension | Time Horizon | NN Controller Structure | Number of Iterations | Runtime (second) | Optimization Setting $[M, N, N_1, N_2, \epsilon, b]$ |
|---|---|---|---|---|---|---|---|
| Simple Car | $\varphi_1$ | 3 | 40 steps | [4,10,2] | 750 | 403.19 | Algorithm 1, b=10 |
| Drone | $\varphi_2$ | 6 | 35 steps | [7,10,3] | 16950 | 354.36 | Algorithm 1, b=20 |
| Quad-rotor | $\varphi_3$ | 12 | 45 steps | [13,20,20,10,4] | 1120 | 6413.3 | $[9, 5, 30, 40, 10^{-5}, 5]$ |
| Multi-agent | $\varphi_4$ | 20 | 60 steps | [21,40,20] | 2532 | 6298.2 | $[12, 5, 30, 1, 10^{-5}, 15]$ |
| Drone & Frame | $\varphi_5$ | 7 | 1500 steps | [8,20,20,10,4] | 84 | 443.45 | $[100, 15, 30, 3, 10^{-5}, 15]$ |
| Dubins car | $\varphi_6$ | 2 | 1000 steps | [3,20,2] | 829 | 3728 | $[200, 5, 60, 3, 10^{-5}, 15]$ |

Table 2: Results on different case studies (details in the appendix)

## 5 CONCLUSION

We introduce STL2LB, a smooth computation graph that lower bounds the robustness degree of an STL specification. We present a neurosymbolic algorithm that uses informative gradients for the design of NN controllers to satisfy STL specifications. We also propose a sampling-based technique to compute robust gradient that does not vanish/explode for long-horizon STL formulas, and provide some strategies to overcome challenges posed by non-differentiable local maxima. We show the efficacy of our training algorithm on a variety of different case studies and present an ablation study that validates the significance of our proposed heuristics.

## 6 REPRODUCIBILITY

The environments used in this paper are standard in the domain of STL controller synthesis. We have provided environment parameters and the hyperparameters used in each of these models. The Appendix sections include sufficient details of our implementation, and our code will be publicly available upon publication.

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

## A   A DETAILED DISCUSSION ON TRAINING ALGORITHM

In this section, we propose the details of the Alg. 2. Although this algorithm is utilized for the case where the gradient may require to be approximated, here we base our reasoning on the assumption that the computed gradient direction increases the objective function in the points of differentiability. However, we evaluate the performance of the algorithm in the presence of gradient approximation, through different challenging case studies.

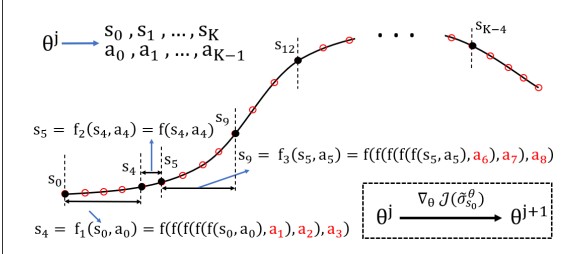

In this algorithm, we initially sample an initial state from the sampled set of initial states $\mathbf{s}_0 \sim \widehat{\mathcal{I}}$. This sampling process can be wise or random. For instance, at every iteration, we may compute the robustness value for all the initial states and select the candidate with the lowest robustness value, or we may sample $\mathbf{s}_0$ uniformly at random.

Figure 2: This figure illustrates the trajectory sampling process for gradient step $j$ in detail. Following the calculation of actions during the forward pass, we maintain dependence between the actions and the state for the sampled time steps. Meanwhile, the remaining actions are set to be independent and fixed. The actions that are fixed are highlighted in red, whereas the dependent actions are denoted in black. The black and red circles represent the sampled and un-sampled time steps.

Here, the initial states are the input data in the training process and the robustness is the objective function. We need to make it certain the objective function is differentiable for a given input data, and this algorithm is designed to do this task. Control synthesis with critical predicate is more efficient than STL2LB but it may result in non-differentiability, and we should re-smooth the function in the point of non-differentiability. This is a challenging task to accurately detect the point of non-differentiability. Thus, we adopt a more conservative approach, and the training algorithm shifts to the smooth version of the robustness function once it is unable to increase the robustness with a critical predicate. To do so, we initialize the parameter use_STL2LB to be False and update it to True in case the robustness is not increased.

Lines [17-22] are proposed to have a more accurate detection for non-differntiable local maxima, where given a small threshold $\epsilon$, keeps the direction of gradient for critical predicate and decreases the learning rate exponentially. In case the gradient of critical predicate is not increasing the robustness for an infinitesimal learning rate, then the chance is high to be in a non-differentiable local maxima.

At the start of the training process, we don't have access to the desired control parameters, but we can envision a desired path for the model to track. This path may not satisfy the temporal specification, but its availability is still valuable information, which its inclusion to the training process can make the problem easier for the Adam optimizer. Therefore, we utilize a desired path and provide a convex and efficient waypoint function and benefit from its gradient in our control synthesis to expedite the training process. We also prioritize the gradient of the waypoint function over the critical predicate. Hence, the lines [14-16] in the algorithm dictates, in case the gradient of this convex function is unable to provide an increment to the robustness value, we should shift to utilize the gradient of critical predicate to increase the robustness function.

In this algorithm, we approximate the gradient for long trajectories through the idea of sampled trajectories visualized in Fig. 2. The idea of sampled trajectories is motivated with the popular idea of (Srivastava et al., 2014) known as drop-out. This approximation may not be precise in some gradient steps of training process. In order to decrease the potential risk, we included a **for**-loop to repeat sampling for time steps and utilize the accumulative gradient to update the parameters. This **for**-loop for computing $\theta_1$ and $\theta_2$ is proposed in lines [7,13] and another **for**-loop for $\theta_3$ is presented in lines [23-29].

## B   SIMPLE CAR: VEHICLE NAVIGATION (SEQUENCE FORMULA)

We assume a 3 dimensional model for the simple car dynamics with pre-specified control inputs, i.e. velocity, $v_k \in [0, 5]$ and steering wheel angle, $\gamma_k \in [-\pi/4, \pi/4]$. Assuming the ZOH discretization with time-step $\delta t = 0.05\,\mathrm{sec}$. This dynamics can be presented as follows:

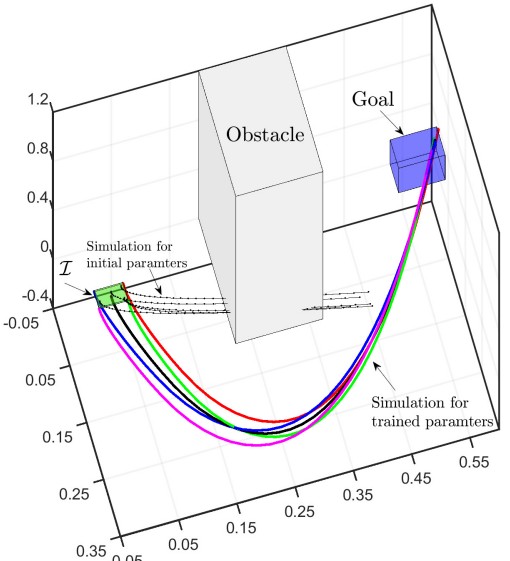 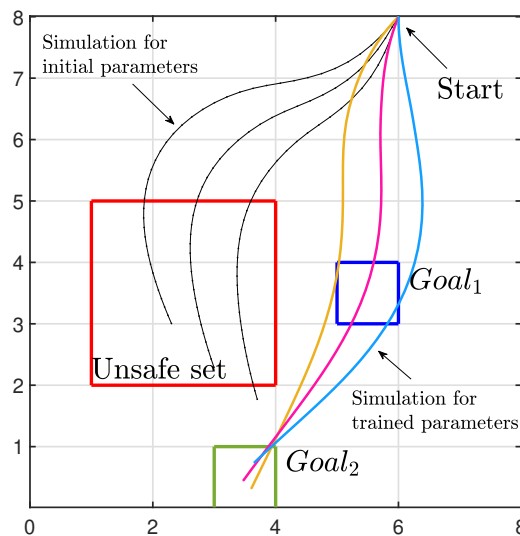

Figure 3: This figure shows the simulated trajectories for trained controller in comparison to the trajectories for naive initial random guess for control parameters. The trajectories are initiated from the set of sampled initial conditions that is the corners of $\mathcal{I}$ with its center.

Figure 4: This figure presents the simulated trajectories from the trained control parameters, on the simple car dynamics, comparing in contrast to the trajectories from initial naive random guess for control parameters. The trajectories are initiated from the set of sampled initial conditions, which is $\theta = \frac{-3\pi}{4}, \frac{-5\pi}{8}, \frac{-\pi}{2}$.

$$\dot{x} = v\cos(\theta), \quad \dot{y} = v\sin(\theta), \quad \dot{\theta} = \frac{v}{L}\tan(\gamma),$$
$$v \leftarrow 2.5\tanh(0.5a_1) + 2.5, \quad \gamma \leftarrow \pi/4\tanh(0.5a_2), \quad a_1, \, a_2 \in \mathbb{R} \tag{7}$$

We plan to train a NN controller with $\mathbf{tanh}()$ activation function and structure $[4, 10, 2]$ for this problem That maps the vector, $[\mathbf{s}_k^\top, k]^\top$ to the unbounded control inputs $[a_1(k), a_2(k)]^\top$. We assume that initially the car is at $(x_0, y_0) = (6, 8)$, but the heading angle can vary in the set $\theta_0 \in [-3\pi/4, -\pi/2]$.

Figure 4 shows the simulation of car's trajectories with our trained controller parameters.

The car is planned to firstly visit $Goal_1$ and once it is there then it should visit $Gaol_2$ and this ordered task should be successfully finished in 40 time steps. However, the car should always avoid the unsafe set according to its assigned task. Unsafe, $Goal_1$ and $Goal_2$ sets are $2D$ sets: $[1, 4] \times [2, 5]$, $[3, 4] \times [0, 1]$ and $[5, 6] \times [3, 4]$, respectively. This temporal task can be formalized in STL framework as follows:

$$\varphi_1 := \mathbf{F}_{[1,40]}\left[Goal_1 \wedge \mathbf{F}[Goal_2]\right] \wedge \mathbf{G}_{[1,40]}\left[\neg \text{ Unsafe set}\right]$$

The black trajectories are the simulation of the initial guess for the controller, which are generated completely at random and are obviously violating the specification. We sampled $\mathcal{I}$ with 3 points ($\theta = -3\pi/4, -5\pi/8, -\pi/2$). We also generated STL2LB with $b = 10$ in $1.31\,\text{sec}$ and starting from mentioned initial guess for control parameters we followed Alg. 1 and trained the parameters within $403.19$ seconds and $750$ gradient steps on CPU with no parallel computing.

## C   DRONE: DRONE MOTION PLANNING (REACH-AVOID FORMULA)

We assume a 6 dimensional model for the quadcopter with pre-specified bounds on the control inputs, $u_1 \in [-0.1, 0.1]$, $u_2 \in [-0.1, 0.1]$, $u_3 \in [7.81, 11.81]$. Assuming the ZOH[8] discretization with time-step $\delta t = 0.05\,\text{sec}$. This dynamics can be presented as follows:

---

[8]Zero Order Hold, fixed control signal over the time step.

$$\dot{x} = v_x, \quad \dot{y} = v_y, \quad \dot{z} = v_z, \quad \dot{v}_x = g\tan(u_1), \quad \dot{v}_y = -g\tan(u_2), \quad \dot{v}_z = g - u_3,$$
$$u_1 \leftarrow 0.1\tanh(0.1a_1), \quad u_2 \leftarrow 0.1\tanh(0.1a_2), \quad u_3 \leftarrow g - 2\tanh(0.1a_3), \quad a_1, a_2, a_3 \in \mathbb{R}.$$
$$(8)$$

We plan to train a NN controller with $\mathbf{tanh}()$ activation function and structure $[7, 10, 3]$ for this problem That maps the vector, $[\mathbf{s}_k^\top, k]^\top$ to the unbounded control inputs $[a_1(k), a_2(k), a_3(k)]^\top$. Here the parameter g = 9.81 is the gravity. We assume that initially the quadcopter is still, $[v_x(0), v_y(0), v_z(0)] = [0, 0, 0]$, and at zero altitude $z_0 = 0$. The initial $x - y$ position of the quadcopter can vary in $[x(0), y(0)] \in [0.02, 0.05] \times [0, 0.05]$.

Figure 3 shows the simulation of quadcopter's trajectories with our trained controller parameters. The quadcopter launches at one point in the set $\mathcal{I}$ and is planned to visit the goal set while avoiding the obstacle. The projection of the obstacle and goal sets into the quadcopter's position states are $[-\infty, 0.17] \times [0.2, 0.35] \times [0, 1.2]$ and $[0.05, 0.1] \times [0.5, 0.58] \times [0.5, 0.7]$, respectively. This temporal task can be formalized in STL framework as,

$$\varphi_2 = \mathbf{G}_{[1,35]}[\neg\,\text{Obstacle}] \wedge \mathbf{F}_{[32,35]}[\text{Goal}]$$

The black trajectories are the simulation of the initial guess for the controller which are generated completely at random and are obviously violating the specification. We sampled $\mathcal{I}$ with 5 points on the corners and center. We also generated STL2LB with $b = 20$ in $0.13\,\text{sec}$ and starting from mentioned initial guess for control parameters we followed Alg. 1 and trained the parameters within 354.36 seconds and 16950 gradient steps on CPU with no parallel computing. In this example, we started from a random initial guess and solved for a control parameter within 6 minutes. However, the authors in (Yaghoubi & Fainekos, 2019) report that they spent 8 hours to find a good initial control parameters. This emphasizes on that fact that our gradients are more informative.

## D QUAD-ROTOR: 12-DIMENSIONAL QUAD-ROTOR (NESTED 3-FUTURE FORMULA)

We assume a 12-dimensional model for the quad-rotor of mass, $m = 1.4$ kg. The distance of rotors from the quad-rotor's center is also $\ell = 0.3273$ meter and the inertia of vehicle is $J_x = J_y = 0.054$ and $J_z = 0.104$ (see (Beard, 2008) for the detail of quad-rotor's dynamics). The controller sends bounded signals $\delta_r, \delta_l, \delta_b, \delta_f \in [0, 1]$ to the right, left, back and front rotors respectively to drive the vehicle. Each rotor is designed such that given the control signal $\delta$ it generates the propeller force of $k_1\delta$ and also exerts the yawing torque $k_2\delta$ into the body of the quad-rotor. We set $k_1 = 3mg/4$ such the net force from all the rotors can not exceed 3 times of its weight, $(g = 9.81)$. We also set $k_2 = 1.5\ell k_1$ to make it certain that the maximum angular velocity in the yaw axis is approximately equivalent to the maximum angular velocity in the pitch and roll axis. We use the sampling time $\delta t = 0.1\,\text{sec}$ in our control process. The dynamics for this vehicle is proposed in equation 9, where $F, \tau_\phi, \tau_\theta, \tau_\psi$ are the net propeller force, pitch torque,

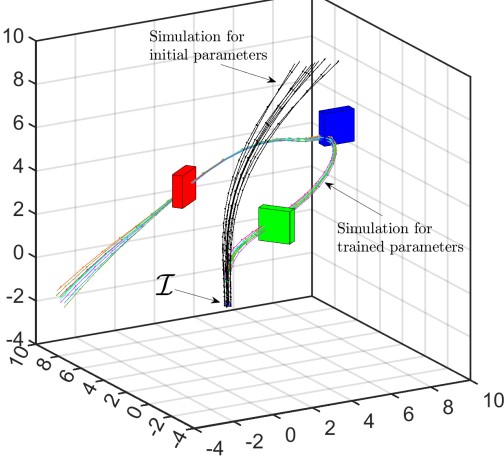

Figure 5: shows the simulation of trained control parameters to satisfy the specified temporal task in companion with the simulation result for initial guess for control parameters.

roll torque and yaw torque respectively. We plan to train a NN controller with $\mathbf{tanh}()$ activation function and structure $[13, 20, 20, 10, 4]$ for this problem that maps the vector, $[\mathbf{s}_k^\top, k]^\top$ to the unbounded control inputs $[a_1(k), a_2(k), a_3(k), a_4(k)]^\top$. In addition to this, the trained controller should be valid for all initial states proposed in equation 9.

Figure 5 shows the simulation of quad-rotor's trajectories with our trained controller parameters. The quad rotor is planned to pass through the green hoop in the next no later than the next 15 time steps and no sooner than the next 10th step and once it passed the green hoop it should pass the blue hoop

$$
\begin{cases}
\dot{x}_1 = \cos(x_8)\cos(x_9)x_4 + (\sin(x_7)\sin(x_8)\cos(x_9) - \cos(x_7)\sin(x_9))x_5 \\
\quad + (\cos(x_7)\sin(x_8)\cos(x_9) + \sin(x_7)\sin(x_9))x_6 \\
\dot{x}_2 = \cos(x_8)\sin(x_9)x_4 + (\sin(x_7)\sin(x_8)\sin(x_9) + \cos(x_7)\cos(x_9))x_5 \\
\quad + (\cos(x_7)\sin(x_8)\sin(x_9) - \sin(x_7)\cos(x_9))x_6 \\
\dot{x}_3 = \sin(x_8)x_4 - \sin(x_7)\cos(x_8)x_5 - \cos(x_7)\cos(x_8)x_6 \\
\dot{x}_4 = x_{12}x_5 - x_{11}x_6 - 9.81\sin(x_8) \\
\dot{x}_5 = x_{10}x_6 - x_{12}x_4 + 9.81\cos(x_8)\sin(x_7) \\
\dot{x}_6 = x_{11}x_4 - x_{10}x_5 + 9.81\cos(x_8)\cos(x_7) - F/m \\
\dot{x}_7 = x_{10} + (\sin(x_7)(\sin(x_8)/\cos(x_8)))x_{11} + (\cos(x_7)(\sin(x_8)/\cos(x_8)))x_{12} \\
\dot{x}_8 = \cos(x_7)x_{11} - \sin(x_7)x_{12} \\
\dot{x}_9 = (\sin(x_7)/\cos(x_8))x_{11} + (\cos(x_7)/\cos(x_8))x_{12} \\
\dot{x}_{10} = -((J_y - J_z)/J_x)x_{11}x_{12} + (1/J_x)\tau_\phi \\
\dot{x}_{11} = ((J_z - J_x)/J_y)x_{10}x_{12} + (1/J_y))\tau_\theta \\
\dot{x}_{12} = (1/J_z)\tau_\psi
\end{cases}
$$

$$
\mathcal{I} = \left\{ \mathbf{s}_0 \;\middle|\; \begin{bmatrix} -0.1 \\ -0.1 \\ -0.1 \end{bmatrix} \leq \begin{bmatrix} x_1(0) \\ x_2(0) \\ x_3(0) \end{bmatrix} \leq \begin{bmatrix} 0.1 \\ 0.1 \\ 0.1 \end{bmatrix} \right\}
$$

$$
\begin{bmatrix} F \\ \tau_\phi \\ \tau_\theta \\ \tau_\psi \end{bmatrix} = \begin{bmatrix} k_1 & k_1 & k_1 & k_1 \\ 0 & -\ell k_1 & 0 & \ell k_1 \\ \ell k_1 & 0 & -\ell k_1 & 0 \\ -k_2 & k_2 & -k_2 & k_2 \end{bmatrix} \begin{bmatrix} \delta_f \\ \delta_r \\ \delta_b \\ \delta_l \end{bmatrix}
$$

$$
\delta_f = 0.5(\tanh(0.5\,a_1) + 1),
$$
$$
\delta_r = 0.5(\tanh(0.5\,a_2) + 1),
$$
$$
\delta_b = 0.5(\tanh(0.5\,a_3) + 1),
$$
$$
\delta_l = 0.5(\tanh(0.5\,a_4) + 1),
$$
$$
a_1,\, a_2,\, a_3,\, a_4 \in \mathbb{R}.
$$

$$(9)$$

in the future 10th to 15th time steps and again once it has passed the blue hoop it should pass the red hoop again in the future next 10 to 15 time steps. This is called a nested future formula, in which we design the controller such that the drone satisfies this specification. This temporal task can be formalized in STL framework as follows:

$$
\varphi_3 = \mathbf{F}_{[10,15]}\big[\, \text{green\_hoop} \,\wedge\, \mathbf{F}_{[10,15]}\big[\, \text{blue\_hoop} \,\wedge\, \mathbf{F}_{[10,15]}\big[\, \text{red\_hoop} \,\big]\,\big]\,\big]
$$

The black trajectories are the simulation of the initial guess for the controller, which are generated completely at random and are obviously violating the specification. We sampled $\mathcal{I}$ with 9 points, that are the corners of, $\mathcal{I}$ including its center. The setting for gradient approximation is $M = 9$, $N = 5$ that implies that for every iteration of backpropagation, we generate a new 9 different set of 5 random time steps $\mathcal{T}^q, q \in [9]$ to sample the trajectory in a way that these sets should satisfy the rule provided in equation 6. We trained the controller with $\bar{\rho} = 0$, in Alg. 2 with optimization setting $(N_1 = 30, N_2 = 40, \epsilon = 10^{-5})$ over 1120 gradient steps (runtime of 6413.3 sec). The runtime to generate STL2LB is also 0.495 sec and we set $b = 5$. The Alg. 2, utilizes gradients from waypoint function, critical predicate, and STL2LB, 515, 544, and 61 times respectively.

## E  MULTI-AGENT: NETWORK OF DUBINS CARS (NESTED FORMULA)

In this example we assume a network of 10 different dubins car that are all under the control of a neural network controller, The dynamics of the system is,

$$
\begin{bmatrix} \dot{x}^i \\ \dot{y}^i \end{bmatrix} = \begin{bmatrix} v^i \cos(\theta^i) \\ v^i \sin(\theta^i) \end{bmatrix}, \quad \begin{array}{l} v^i \leftarrow \tanh(0.5a_1^i) + 1, a_1^i \in \mathbb{R} \\ \theta^i \leftarrow a_2^i \in \mathbb{R}, \ i \in [10]. \end{array} \tag{10}
$$

which is a 20 dimensional multi-agent system with 20 controllers, $v^i \in [0,1], \theta^i \in \mathbb{R}, \ i \in [10]$. Figure 6 shows the initial position of each dubins car in $\mathbb{R}^2$ in companion with their corresponding Goal sets. The cars should be driven to their goal sets, and they should also keep a minimum distance of $d = 0.5$ meters from each other while they are moving toward their goal sets. We assume a sampling time of $\delta t = 0.26$ for this model, and we plan to train a

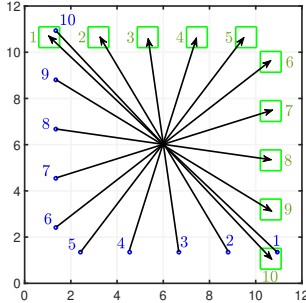

Figure 6: Shows the agents vs goal sets

NN controller with $\mathbf{tanh}()$ activation function and structure $[21, 40, 20]$ via Alg. 2, for this problem that maps the vector, $[\mathbf{s}_k^\top, k]^\top$ to the unbounded control inputs $\{a_1^i(k), a_2^i(k)\}_{i=1}^{10}$. This temporal task can be formalized in STL framework as follows:

$$
\varphi_4 := \left( \bigwedge_{i=1}^{10} \mathbf{F}_{[20,48]}\Big[\, \mathbf{G}_{[0,12]}\big[\big(x^i(k), y^i(k)\big) \in \text{Goal}^i\big]\,\Big] \right) \bigwedge
$$

$$
\left( \bigwedge_{\substack{i \neq j \\ i,j \in [10]}} \mathbf{G}_{[0,60]}\Big[\big(\mid x^i(k) - x^j(k) \mid > d\big) \vee \big(\mid y^i(k) - y^j(k) \mid > d\big)\Big] \right)
$$

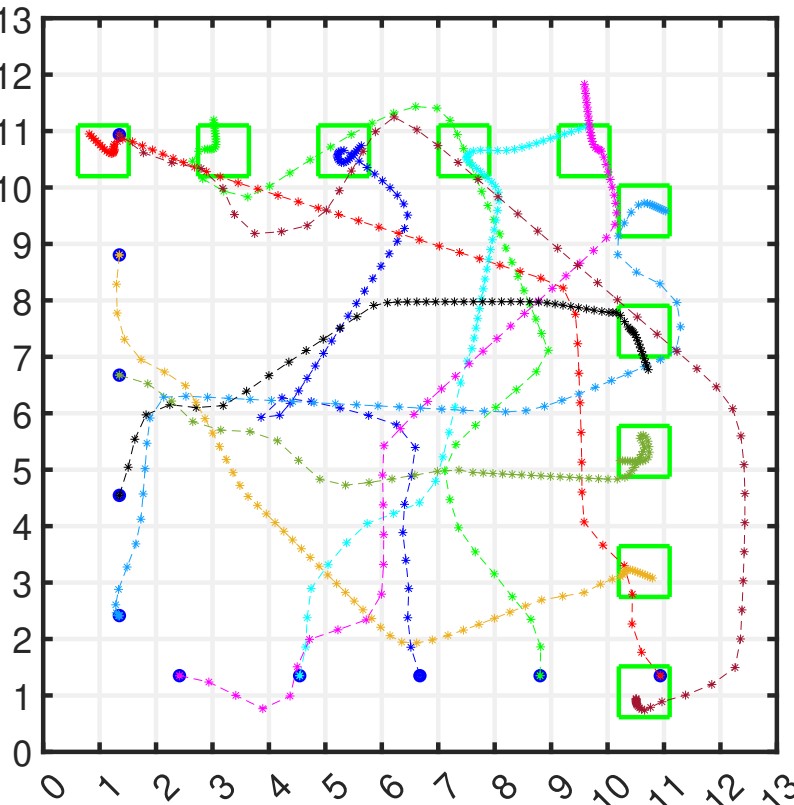

Figure 8: Shows the simulation of trained controller on the multi-agent system of 10 connected Dubins cars. The cars start from an assigned initial position and follow the command of a central NN controller, which we have trained with Alg. 2. This controller makes it certain that cars arrive to and stay in their Goal sets based on the specification and will always keep a pre-specified distance from each other over the course of traveling. The trajectories are intentionally plotted with astric points to spot the position of cars at every single time step. The identity of each agent and its assigned Goal sets is also available in Figure 6. Our observation shows that the agents finish their personal tasks (First component of $\varphi_4$) in different times.

Figure 8 shows the simulation of the trajectories for the trained controller, and Figure 7 presents the simulation of trajectories for the initial guess for control parameters. We observe that our controller manages the agents to satisfy the task in different times. We present the time steps with astric to make a more clear presentation of the task. Due to the high dimension and complexity of the task in this example, we were unable to solve it with Alg. 1, but we were able to solve this with Alg. 2 within a 6298 seconds and 2532 gradient steps. We also set the optimization setting as, $M = 12, N = 5, N_1 = 30, N_2 = 1, \epsilon = 10^{-5}, b = 15$. The runtime to generate STL2LB is also 6.2 seconds. Over the course of the training process we utilized 187, 1647 and 698 gradients from way point function, critical predicate and STL2LB respectively.

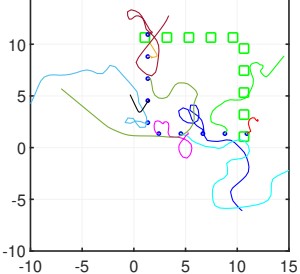

Figure 7: Shows the simulation of trajectories for the initial guess of the control parameters.

## F    DRONE & FRAME: LANDING A DRONE

We again use the 6 dimensional dynamics for the drone presented in equation 8. The horizon of the temporal task is 1500 time steps with $\delta t = 0.05$ sec. The drone launches at a helipad located at $(x(0), y(0), z(0)) = (-40, 0, 0)$. We also accept a deviation of 0.1 for $(x(0)$ and $y(0))$ and we train the controller to be valid for all the states sampled from this region. The helipad is also 40 meters far from a building located at $(0, 0, 0)$. The building is 30 meters high, where the building's footprint is $10 \times 10$ meters. We have also a moving platform with dimension $2 \times 2 \times 0.1$ that is starting to move from $(10, 0, 0)$ with a variable velocity, modeled as, $\dot{x}^f = u_4..$ We also accept a deviation 0.1 for, $x^f(0)$ and our trained controller is also robust with respect to this deviation. We define $\widehat{\mathcal{I}}$ with 9 samples that are located on corners of, $\mathcal{I}$ including its center.

The frame is also required to keep always a minimum distance of 4.5 meters from the building. We train a NN controller that controls the drone and the platform together such that the drone will land on

the platform with relative velocity of at most $1\ m/s$ on $x, y$ and $z$ directions and its relative distance is also at most 1 meter in $x, y$ direction and $0.4$ meter in $z$ direction. This temporal task can be formulated as a reach-avoid formula in STL as follows:

$$\varphi_5 = \begin{array}{l} \mathbf{G}_{[0,1500]} [\neg\text{obstacle}] \\ \land\ \mathbf{F}_{[1100,1500]}[\text{Goal}] \\ \land\ \mathbf{G}_{[0,1500]}[x^f(k) > 9.5] \end{array} \quad,\quad \text{Goal}=\left\{ \begin{bmatrix} x(k) \\ y(k) \\ z(k) \\ v_x(k) \\ v_y(k) \\ v_z(k) \\ x^f(k) \end{bmatrix} \ \Bigg|\ \begin{bmatrix} -1 \\ -1 \\ 0.11 \\ 0 \\ -1 \\ -1 \end{bmatrix} \leq \begin{bmatrix} x(k) - x^f(k) \\ y(k) \\ z(k) \\ v_x(k) \\ v_y(k) \\ v_z(k) \end{bmatrix} \leq \begin{bmatrix} 1 \\ 1 \\ 0.6 \\ 2 \\ 1 \\ 1 \end{bmatrix} \right\}$$

We plot the simulated trajectory for the center of set of initial states $\mathcal{I}$, in Figure 9. The NN controller's structure is specified as $[8, 20, 20, 10, 4]$ and uses the $\tanh$ activation function. We initialize it with a random guess for its parameters. The simulated trajectory for initial guess of parameters is also depicted in black. The setting for gradient approximation is $M = 100$, $N = 15$ that implies that for every iteration of backpropagation, we generate a new 100 different set of 15 random time steps $\mathcal{T}^q, q \in [100]$ to sample the trajectory, such that these sets satisfy the specified requirements in equation 6. We trained the controller with $\bar{\rho} = 0$, over 84 gradient steps (runtime of $443$ sec). The runtime to generate

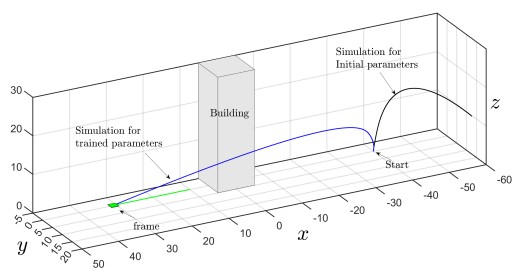

Figure 9: This figure shows the simulated trajectory for trained controller in comparison to the trajectories for naive initial random guess. The frame is moving with a velocity determined with the controller that also controls the drone.

STL2LB is also $7.74$ sec and we set $b = 15$. In total, the Alg. 2, utilizes gradients from waypoint function, critical predicate, and STL2LB , $5, 71$, and $8$ times respectively.

## G  ANALYSIS FOR DIFFERENT MODULES INCLUDED IN ALGORITHM 2

In this section, we perform an analysis over the three modules proposed in Alg. 2. We focus on the landing drone mission and compare the results once a module is disabled from the algorithm. In the first step, we remove the way point function from Alg. 2 and show the performance of algorithm with critical predicate based sampling process and safe-resmoothing. In the next step, we also dis-regard the presence of critical time in time sampling and train the controller with completely at random time sampling. This implies we solve the problem relying only on safe-resmoothing module. Table 3 shows the efficiency of training process in each case and Fig. 10 compare the learning curves. Our experimental result shows, the control synthesis for drone landing mission faces a small reduction in efficiency when the way point function is disregarded and fails when the critical predicate is also removed from time sampling.

## H  DUBINS CAR: GROWING TASK HORIZON FOR DUBINS CAR

In this example, we utilize the dubins car to provide an experiment that shows us, (a) the advantage for reformulating the Alg. 1 to the Alg. 2 and (b) the advantage for inclusion of sampling based gradient approximation. To that end, given a scale factor $a > 0$ and a time horizon $K$. We plan to train a neural network controller with structure $[3, 20, 2]$ with $\tanh()$ activation function and a pre-defined initial guess for control parameters to drive a dubins car, with dynamics,

$$\begin{bmatrix} \dot{x} \\ \dot{y} \end{bmatrix} = \begin{bmatrix} v\cos(\theta) \\ v\sin(\theta) \end{bmatrix}, \quad \begin{array}{l} v \leftarrow \tanh(0.5a_1) + 1, a_1 \in \mathbb{R} \\ \theta \leftarrow a_2 \in \mathbb{R} \end{array}$$

to satisfy the following temporal task,

$$\varphi_6 := \mathbf{F}_{[0.9K,K]}[\text{Goal}] \land \mathbf{G}_{[0,K]}[\neg\text{Obstacle}].$$

The dubins car starts from $(x(0), y(0)) = (0, 0)$. The obstacle is also a square centered on $a/2$ with the side length, $2a/5$ and the Goal region is again a square centered at $9a/10$ with the side length

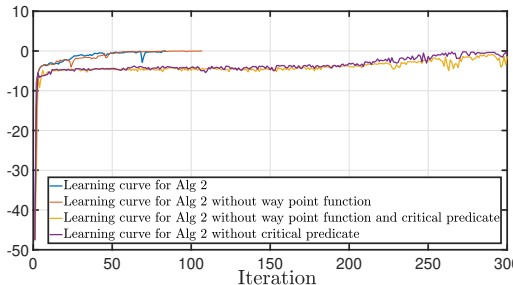

Figure 10: Shows the learning curve for training processes. This figure shows the Alg. 2 concludes successfully in 84 iterations while removing the way point it concludes in 107 iterations. The algorithm also fails if the critical predicate is not considered in time sampling.

| Way-point function | Critical Predicate | Safe Resmoothing | Number of Iterations | Runtime (seconds) |
|:---:|:---:|:---:|:---:|:---:|
| ✓ | ✓ | ✓ | 84 | 443 |
| × | ✓ | ✓ | 107 | 607 |
| ✓ | × | ✓ | NF$[-0.74]$ | 6971 |
| × | × | ✓ | NF$[-1.32]$ | 4822 |

Table 3: Shows the numerical results for the training algorithms. In case the training process does not provide positive robustness within 300 gradient step we report it with NF[.] which indicates the value of robustness in iteration 300. In this table we disable the main modules in Alg. 2 step by step and report the extent of reduction in efficiency. The symbol ✓ indicates the module is included and × indicates the module is neglected.

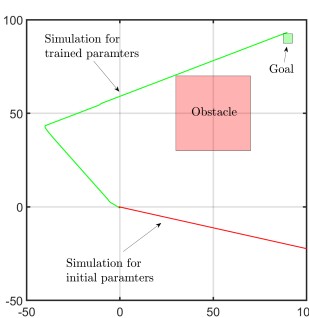

Figure 11: Show the simulation of the results for Dubins car. One of the main issues of using NN controller for this problem is that the $\tanh()$ activations functions gets saturated over long horizon and the straight lines confirm this. However, Alg. 2 has successfully solved for this long horizon task in the presence of this source of difficulty. Consider a) the vehicle should visit the Goal set after $k = 900$, b) the controller gets saturated at some time steps before that, and c) the Goal set may not be reachable before $k = 900$. Therefore, the algorithm is required to firstly locate the car in a specific zone before controller's activation functions are saturated and then drive it toward the Goal set with a saturated controller, or in another word, constant velocity and fixed heading angle.

$a/20$. We solve this problem for $K = 10, 50, 100, 500, 1000$ and we also utilize $a = K/10$ for each case study. We solve each case study with both Algorithms 1 and 2 and for each algorithm we consider two different conditions, where the former is to utilize the real gradient and the latter is to employ the approximated gradient using the time sampling technique. Consider we keep the initial guess and the controller's structure fixed for all the training processes, and we also manually stop the process once the number of iterations exceeds 8000 gradient steps. We also assume a singleton as the set of initial states $(0, 0)$ to present a more clear comparison. The runtime and the number of iterations for each training process is presented in table 1. Figure 11 shows the simulation trajectories trained from Alg. 2 with gradient approximation for $K = 1000$ time steps in companion with the simulation of trajectories for the initial guess of controller parameters.

## I  VERIFICATION OF RESULTS

In order to verify the results, we adopt the approach presented in (Lindemann et al., 2022). To that end, we set the probabilities $\delta = \%0.1, \varepsilon = \%99.9$ and prove:

$$\Pr\left[\Pr[-\rho(\varphi, \sigma_{\mathbf{s}_0}^\theta, 0) \leq -\bar{\rho}] \geq \varepsilon\right] \geq 1 - \delta \tag{11}$$

Based on the methodology proposed in (Lindemann et al., 2022) we are required to generate a minimum of $3.8005 \times 10^6$ random trajectories to formalize the statement equation 11. Accordingly, we sampled $4 \times 10^6$ initial states $\mathbf{s}_0 \in \mathcal{I}$ uniformly at random and simulated the corresponding trajectories for all the examples, and we faced no counter example with negative robustness value.

## J  PROOF OF LEMMA 1

Assume a STL specification $\varphi$ which defines a set of predicates on the trajectory states in $\sigma_{\mathbf{s}_0}$ and establishes a positive normal form relation over these predicates (see equation 3). Let's denote the set of quantitative semantics for these predicates with $\mathcal{A}$,

$$\mathcal{A} = \{a_1, a_2, \cdots, a_N\}.$$

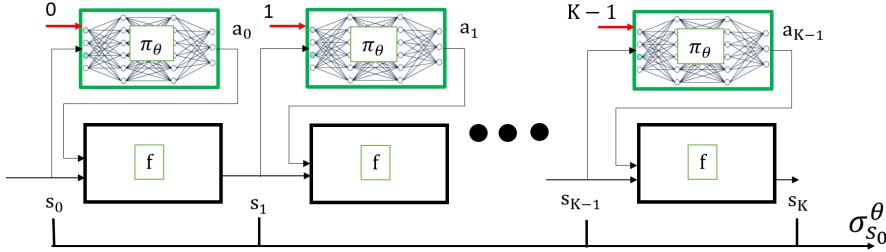

Figure 12: Shows an illustration of the recurrent structure for the control feedback system.

The non-smooth robustness function $\rho(\sigma_{\mathbf{s}_0}, \varphi, 0)$ can be considered a recursive combination of $\min()/\max()$ operators over the quantitative semantics $a_i \in \mathcal{A}, i \in [N]$. This implies, in case we replace the $\min()/\max()$ operators with their guaranteed lower-bounds, the robustness value will likewise be under-approximated, provided that the formula $\varphi$ is in positive normal form.

Let's use the notation $\mathsf{max2}()$ and $\mathsf{min2}()$ to refer to the specific case of $\max()$ and $\min()$ applied to two quantities. Then the maximum or minimum for a multitude of quantities is also computable recursively over the functions $\mathsf{max2}()/\mathsf{min2}()$. As an example for a case of 5 quantities, $x_1, x_2, x_3, x_4, x_5$, we can write:

$$\max(x_1, x_2, x_3, x_4, x_5) = \mathsf{max2}\,(\,\mathsf{max2}\,(\,\mathsf{max2}\,(x_1\,,\,x_2)\,,\,\mathsf{max2}\,(x_3\,,\,x_4))\,,\,x_5)$$

This implies in case we replace the functions $\mathsf{max2}()/\mathsf{min2}()$ with a guaranteed lower-bound, the functions $\min()/\max()$ will be under approximated and thus the robustness function is also under-approximated.

We know for all $x, y \in \mathbb{R}$, $\mathsf{max2}(x, y) = y + \mathsf{ReLU}(x - y)$ and $\mathsf{min2}(x, y) = x - \mathsf{ReLU}(x - y)$. We also know, for all $z \in \mathbb{R}$, $\mathsf{swish}(z) < \mathsf{ReLU}(z)$ and $\mathsf{softplus}(z) > \mathsf{ReLU}(z)$ where,

$$\mathsf{softplus}(z) = \frac{1}{b} \log\left(1 + e^{bz}\right), \quad \mathsf{swish}(z) = \frac{z}{1 + e^{-bz}}, \qquad b > 0.$$

This motivates us to define the smooth functions $\mathsf{max2s}()/\mathsf{min2s}()$ as follows:

$$\mathsf{max2s}(x, y) = y + \mathsf{swish}(x - y), \quad \mathsf{min2s}(x, y) = x - \mathsf{softplus}(x - y)$$

and claim for all $x, y \in \mathbb{R}$, $\mathsf{max2s}(x, y) < \mathsf{max2}(x, y)$ and $\mathsf{min2s}(x, y) < \mathsf{min2}(x, y)$. Therefore, considering robustness function $\rho(\sigma_{\mathbf{s}_0}, \varphi, 0)$, replacement of $\mathsf{max2}()$ with $\mathsf{max2s}()$ and $\mathsf{min2}()$ with $\mathsf{min2s}()$ results in a smooth lower-bound for robustness function that is a recursive combination of $\mathsf{max2s}()/\mathsf{min2s}()$ over the elements of set $\mathcal{A}$. Since $\mathsf{STL2NN}(\sigma_{\mathbf{s}_0})$ is exactly equivalent to $\rho(\sigma_{\mathbf{s}_0}, \varphi, 0)$, we denote the proposed smooth robustness with $\mathsf{STL2LB}(\sigma_{\mathbf{s}_0}\,; b)$ and conclude,

$$\forall (\sigma_{\mathbf{s}_0}, b) \in \mathbb{R}^{nK} \times \mathbb{R} \;:\; \mathsf{STL2LB}(\sigma_{\mathbf{s}_0}\,; b) \leq \mathsf{STL2NN}(\sigma_{\mathbf{s}_0})$$

## K    AN ILLUSTRATION FOR THE RECURRENT STRUCTURE

Figure 12 shows an illustration for computation graph of trajectory states in our control feedback system. The controller is a feed forward neural network that receives the time and state and returns a decision. The parameters of this controller will be trained to satisfy a temporal property, formulated in STL framework. As the control parameters are repeated over the computation graph, this recurrent structure is similar to RNN structure and also suffers from vanishing/exploding gradient for a long horizon trajectory.

## L    COMPARISON BETWEEN THE PERFORMANCE OF STL2LB AND OTHER SMOOTH SEMANTICS IN TRAINING.

Considering the Boltzmann operator and log-exp-sum operators provided in equation 5 and the STL robustness proposed in equation 3, the methodology proposed in (Pant et al., 2017) suggests to replace

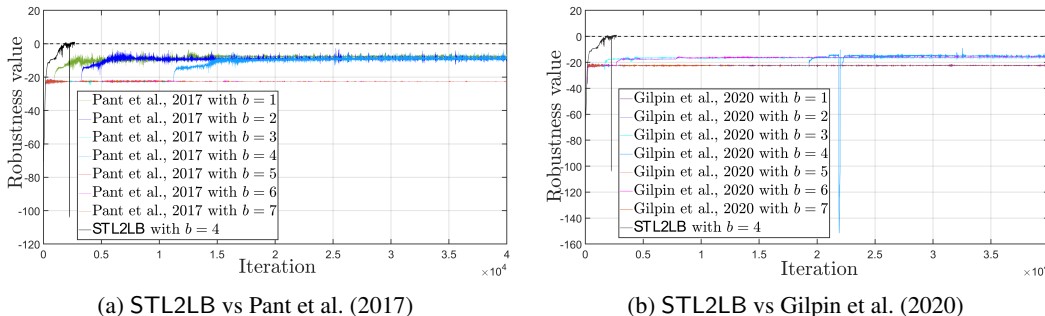

(a) STL2LB vs Pant et al. (2017)  (b) STL2LB vs Gilpin et al. (2020)

Figure 14: Shows the comparison of learning curve between STL2LB with the smooth semantics proposed by Pant et al. (2017) and Gilpin et al. (2020). This figure shows STL2LB trains the controller in 2700 gradient steps and provide positive robustness. This is, while the smooth semantics from Pant et al. (2017) and Gilpin et al. (2020) are unable to provide positive robustness within up to 40000 gradient steps. Since the operational zone is wide, the proposed techniques in Pant et al. (2017) and Gilpin et al. (2020), do not accept $b \geq 8$ due to computational errors proposed in Example 1. Although STL2LB have no limit for handling large values of hyper-paramter $b$, here we also restrict STL2LB for $b = 4$ to have more clear comparison.

$\max()/\min()$ operators in equation 3 with log-exp-sum smooth operator. On the other hand, the methodology proposed in (Gilpin et al., 2020) suggests to replace $\max()$ operator with Boltzmann operator and $\min()$ operator with log-exp-sum. However, both of these techniques suffer from the computational issues proposed in Example 1. This problem manifest when the quantitative semantics for predicates is sufficiently large. Therefore, we intentionally assume a wide operational zone for a Dubin's car and show including smooth semantics from Pant et al. (2017) and Gilpin et al. (2020) into training process results in failure, while incorporation of STL2LB results in an efficient training process. In this training process, we assume the sampling time is $\delta t = 2\,\text{sec}$, the velocity of car, is constrained with $v \in [0, 2]$ and the set of initial states in $\mathcal{I} = [0, 2] \times [0, 2]$. Dubin's car is also planned to satisfy the STL specification:

$$\varphi_7 = \mathbf{G}_{[0,35]}[\neg\text{Unsafe}] \bigwedge \mathbf{F}_{[12,17]}[\text{Goal}_1] \bigwedge \mathbf{F}_{[30,35]}[\text{Goal}_2]$$

where the sets $\text{Unsafe}, \text{Goal}_1, \text{Goal}_2$ are shown in Fig. 13. The controller is also feedforward neural network with structure $[3, 10, 2]$. Since the operational zone is wide, the quantitative semantics of predicates are large to the extent that smooth semantics from Pant et al. (2017) and Gilpin et al. (2020) can not handle $b \geq 8$ and thus we are restricted to utilize $b < 8$. On the other hand, STL2LB does not face numerical problems mentioned in Example 1. Therefore, we plan to show in this condition, unlike Pant et al. (2017) and Gilpin et al. (2020), STL2LB do not face this issue. To validate this issue, we train for controller with STL2LB and the other two techniques. Fig. 14 shows the results of these training processes, and clearly shows STL2LB is the only smooth semantics that can be utilized to solve this problem. In this case study, $b = 4$ provides the best solution for Pant et al. (2017) and Gilpin et al. (2020) thus we also generate STL2LB with $b = 4$ to have a more clear comparison [9]. We also compare the simulated trajectories for this case with the result of STL2LB in Fig. 13.

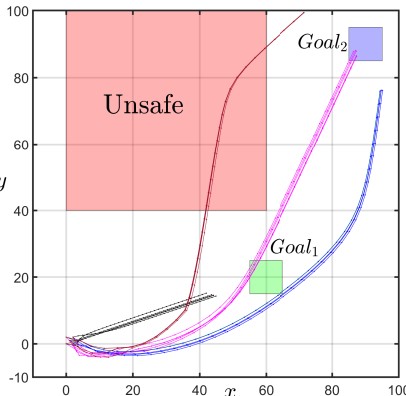

Figure 13: shows the simulation of trained control parameters to satisfy the specified temporal task in companion with the simulation result for initial guess for control parameters. The black, blue, magenta and red trajectories are for the initial guess, Pant et al. (2017), STL2LB and Gilpin et al. (2020), respectively. The only trajectory that satisfies the specification is for STL2LB.

All the three training algorithms are identical except the smooth semantics that is utilized. They are also started from a unique guess for initial control parameters. We utilized the MATLAB function adamupdate(), that automatically tunes the learning rate and momentum in SGD.

---

[9]The low value for $b$ is the main reason behind the sudden drop in the learning curve for STL2LB. We can effectively compensate for this issue by increasing the value of $b$. Here, increasing to $b = 15$ is sufficient.

