# OpenReview forum: "Scaling Safe Learning-based Control to  Long-Horizon Temporal Tasks"
_ICLR.cc/2024/Conference — Submitted to ICLR 2024_

### Official Review · Reviewer_NFTL · 2023-10-21

**Soundness:** 3 good
**Presentation:** 2 fair
**Contribution:** 2 fair
**Rating:** 3
**Confidence:** 4

**Summary:**

This paper studies control problems in which a neural network policy must satisfy a signal temporal logical formula on a long time horizon. The specific contributions are twofold: (1) a differentiable technique for evaluating the temporal logic formula, and (2) a lower-complexity approach to evaluating gradients which also reduces the chances of exploding gradient issues. Simulation experiments illustrate the relative impact of these two techniques.

**Strengths:**

- The problem of long-horizon planning is certainly challenging and important, and there is a gap in the literature here with regard to learning to satisfy temporal logic specifications.
- The proposed technique for smoothing the specification is intuitive and easy to follow. (The same cannot be said of the gradient approximation scheme.)
- The proposed approaches appear to handle the tested scenarios well, and tradeoffs are readily apparent and well explained.

**Weaknesses:**

- nit: there are quite a few sporadic syntax and grammar issues, e.g. in the first paragraph there are two instances of improper spacing near punctuation marks. Similar issues around references: you may consider using the \citep{} option for parenthetical citations.
- In Section 3, the paper clearly states that a condition must be satisfied for all initial states, yet the training objective is an expectation measured at only a finite number of samples. The appendix contains an abrupt mention of this point which leads me to believe there is more going on here, but it is a fairly serious omission from the main text and is also not really clarified in the appendix.
- In “Challenge 1” and elsewhere, it is asserted that existing frameworks for stochastic optimization cannot handle non-smooth objectives. This is patently false: every ReLU network ever has been non-smooth, and yet SGD/Adam/… seem to do just fine. Obviously, there is also a rich theory in non-smooth optimization, sub-gradient methods, etc. My objection here is mainly that the paper just asserts that smoothness is critical without ever supporting that claim.
- Where is the proof of Lemma 1? It seems important, but is nowhere to be found. Same for the equation at the bottom of page 4.
- The modified swish and soft plus functions below Ex. 1 appear to be non-smooth, and what smoothness there is derives only from numerical precision issues. This is less than satisfying. Surely if we are hitting numerical precision issues that points to something more subtle going on, right? For example, it is well-documented in the literature that unstable closed loop dynamics yield exploding gradients in these kinds of policy optimization problems.
- What is the parameter \bar\rho in Alg. 1? (nit: note also that “Algo” is not a common abbreviation for Algorithm. I believe the IEEE standard, at at least, is “Alg.”)
- I do not follow the entire discussion of the gradient approximation scheme. Fig. 1 makes sense to me and I follow that part of the discussion, but my concern is that the sampling discussion following Definition 2 (and especially the part to the right of Alg. 2) is completely uninterpretable to me. A couple direct questions:
    - Is the matrix S actually being approximated at specific rows, or are entire rows being left out?
    - How is it more efficient to compute the gradient at a sampled point? Doesn’t this essentially require backpropagating through all time steps from the end of time to the beginning, regardless of whether or not you are going to then throw away some of the gradient information?
- Please help me to understand the last sentence before Section 4. Didn’t the guarantees come from simply evaluating the (potentially smoothed) STL formula from every initial condition—i.e., the optimization objective? Why does it matter if we change the algorithm used to approximate gradients?
- nit: "dubins" should be capitalized in the first paragraph of section 4.
- The authors point to quite a lot of closely related work in this space: I feel that the experiments should benchmark the proposed approaches against one or two of these recent methods. The “comparison” paragraph on page 8 alludes to one such comparison, but it seems like a straw man because of the radical difference in performance. If this is not a straw man comparison, the paper should do a much clearer job of establishing why the baseline is a strong baseline.
    - Relatedly, I would be interested to see how much of a difference the smoothed STL formula makes in learning (essentially an ablation of the first contribution of this paper).

**Questions:**

please see above

---

> ### Author Response · Authors · 2023-11-17
> **Response to Reviewer NFTL [First part]**
>
> We Thank for all of your valuable comments and suggestions. Here we answer to your comments titled as "Weakness" and "Question". And we start with "Weakness" section.
>
> Weakness:
>
> Weakness 1 : We will carefully review the language and polish it, we will also use \citep{} for citations.
>
> Weakness 2 : You are right. The problem that we state is the ultimate goal for our training algorithm, but in the main body of the paper, we do not address this problem. However, we added a footnote (see footnote 3 in the updated text document) to clarify this point for the reader. We need to add, there are prior techniques in the literature (such as [1]) which can give us provable statistical guarantees. We refer you to Appendix H, which applies the technique from [1] to give probabilistic verification guarantees. We note that we still use the relaxed training objective (expectation of robustness values over initial conditions rather than guaranteeing that the min robustness value over all initial conditions is positive). We choose the relaxed condition because ensuring the stricter condition is intractable for both classical control techniques (due to highly nonlinear environments) and neural network controllers (due to lack of guaranteed optimization algorithms).
>
> Weakness 3 : You are absolutely right, this was poor writing on our part. What we meant to say is that in the context of the robustness of STL specifications, non-smoothness becomes a deterrent. For example, previous work in [3] compares smooth STL semantics with MILP based formulations that use non-smooth, min-max based STL semantics, and shows considerable advantages in using smooth semantics. Thus, we assume community knowledge of sub-optimality in using non-smooth STL semantics for controller synthesis to justify our use of smooth semantics that are guaranteed lower bounds on the actual robustness.
>
> Weakness 4 :   We omitted a formal proof due to lack of space, but hoped to give insight into the proof in the paragraphs preceding the Lemma in Section 3.1. Here’s a brief proof sketch:
> STL2NN is syntactically equivalent to the robustness function in [2] as it is simply a reformulation of min/max operations in the robustness function as ReLU units. The main idea here is that:
> \\[
> 	\\min(a_1 , a_2) =  a_1 - ReLU(a_1-a_2)    \\quad(i.a)  \\quad \\& \quad \\ max(a_1 , a_2) = a_2 + ReLU(a_1-a_2)   \\quad(i.b)
> \\]
> Next, we can show that
> \\[
> \\forall x: \\quad  \\text{swish}(x) < ReLU(x)  \\quad(ii.a) \\quad \\& \\quad  \\forall x:  \\quad \\text{soft-plus}(x) > ReLU(x)  \\quad(ii.b)
> \\]
>
> This follows from standard nonlinear inequalities about log/exp functions. For example, here’s the proof for (ii.a) by doing case analysis:
> Note that $ ReLU(x) = \\max(x,0)$ and $\\text{swish}(x) = x/(1+e^{-bx}), \\quad b>0$.
>
> Case I: $ x > 0  \\quad \\implies \\ max(x,0) = x$. Since, $\\forall x \\in \\mathbb{R}:\quad  e^{-bx} > 0$, we conclude $ (1+e^{-bx}) > 1$, so $x/(1+e^{-bx}) < x$
>
> Case II: $x \\leq 0 \\quad \\implies \\ max(x,0) = 0$, while $\\text{swish}(x)$ is a negative number. <completing the proof>
>
> We can similarly show that $\\forall x \\in \\mathbb{R} \\quad  \\text{soft-plus}(x) > ReLU(x)$
>
> We can use (ii.a) and (ii.b) to show that replacing ReLU in (i.a) by soft-plus and ReLU in (i.b) by swish guarantees that the RHS expressions are lower-bounds on min() and max() respectively.
>
> To see this, let’s define the smooth functions  $\\mathsf{mins}()$ and $\\mathsf{maxs}()$  as follows:
> \\[
>   \\mathsf{mins}(a_1 , a_2) =  a_1 - \\text{soft-plus}(a_1-a_2) , \\quad (iii.a) \\quad \\& \\quad   \\mathsf{maxs}(a_1, a_2) = a_2 + \\text{swish}(a_1-a_2) \\quad (iii.b)
> \\]
> From (ii.a), (ii.b) we can see that
> \\[
> \\min(a_1 , a_2) >  \\mathsf{mins}(a_1,a_2) \\quad (iv-a)  \\quad \\& \\quad   \\max(a_1 , a_2) > \\mathsf{maxs}(a_1,a_2) \\quad (iv-b).
> \\]
> Rest of the proof follows by structural induction on the syntax tree of a given STL formula. Essentially, the robustness computation of an STL formula requires the robustness values of all subformulas over all times in the relevant time-scope of the formula. We can show that for all STL operators (i.e. $\wedge, \vee$, $\mathbf{F}$, $\mathbf{G}$, $\mathbf{U}$, $\mathbf{R}$), the robustness computation is essentially a $\\min()$ or $\\max()$ over robustness values of sub-formulas. Since robustness computation using $\\mathsf{mins}()$ and $\\mathsf{maxs}()$ is a lower bound for the actual $\\min()/\\max()$ operation, the robustness computation operations using $\\mathsf{mins}()/\\mathsf{maxs}()$ semantics also lower bound the robustness of the actual formula.
>
> We also added a proof sketch to the updated text document, please see Appendix J.

---

> ### Author Response · Authors · 2023-11-17
> **Response to Reviewer NFTL [Second part]**
>
> Weakness 5: You are right that these functions are not smooth. However, please note that even a smooth function will face numerical precision issues due to fundamental limits on the smallest number that can be represented in any modern computer. Both Matlab and Python have a pre-specified tolerance for numerical precision, and they do not recognize numbers less than this precision. As we have shown in our response to reviewer 1, by picking a value (720) for the hyper-parameter $\tau$, we can make the difference in the derivative of the modified swish and soft-plus functions with their smooth form, as small as $10^{-310}$. Increasing $\tau$ will make the difference even smaller. The numerical precision for Python/Matlab for “double” precision numbers is of the order of $10^{-308}$ (we selected $\\tau=50$ in our experiments). Thus, we argue that from a computational perspective, the functions can be practically smooth. We also remark that the advantage offered by the modified swish/soft-plus functions outweighs any concerns about computational difficulties when compared to using log-exp-sum or the Boltzmann operator. The latter operators make it impossible to compare both large and very small numbers due to numerical issues and thus impossible to compute smooth versions of the min/max operations. For example, log-exp-sum can not find the maximum of 80 and 81 if b=10,
> \\[
> \\text{log-exp-sum}: \\quad \\widetilde{max}(80,81) = \\frac{1}{10} \\log (\\exp(10 \times 80)+ \\exp(10 \times 81))
> \\]
> as you see $\\exp(800)$ and $\\exp(810)$ are not computable in MATLAB.
>
>
> Weakness 6 :  We will revise the abbreviation for Algorithm. The parameter $\\bar{\\rho}$ serves as a lower threshold for the required robustness that the controller must attain. When $\\bar{\\rho}=0$, the controller is trained solely to satisfy the given temporal specification. Alternatively, setting $\\bar{\\rho} > 0$ allows us to enhance safety levels by achieving a robustness greater than zero.
>
> Weakness 7:  Please see the next part (Response to Reviewer NFTL [Third part]), This response is provided after references.
>
> Weakness 8:  You are correct – this is a good catch. It is true that guarantees indeed come from evaluating the smoothed STL formula. What we wanted to say is that the re-smoothing process may decrease the value of STL2LB. The sentence as written was incorrect/confusing, and we have removed it from the updated text document.
>
> Weakness 9: We have fixed this.
>
> Weakness 10 :  There are two main messages that we want to convey in this paper: (1) it is important for the smooth robustness semantics of STL to be rigorous lower-bounds for the actual robustness, (2) synthesizing neural network based controllers for long horizon temporal tasks faces problems of vanishing and exploding gradients, and to make learning-based synthesis feasible, we need better algorithms to approximate the gradient.
>
> The related work in this space is along two orthogonal directions: (1) the kind of smooth approximations to min/max used to give smooth semantics for STL robustness, (2)  the type of control policy obtained by the synthesis algorithm. Most of the related work along the second direction, focuses on generating {\\em open loop} control policies (i.e. directly synthesizing the actions). It is well-known that open loop policies are not robust to noise, hence we prefer closed-loop feedback controllers. The only work that synthesizes closed-loop feedback control is a baseline that can lead to an apples-to-apples comparison. The comparison to open-loop policy synthesis is unfair because the complexity of generating open-loop policies is much lesser.
>
> The other dimension is about smooth approximations of min/max. Superficially, the computational burden using STL2LB and other smooth semantics may look similar. However, during training, if robustness values of the signal predicates become high or low, then other smooth min/max versions give incorrect answers, potentially giving nonsensical values for the gradient. This is a serious computational issue.  You can see example 1 in the paper how the gradient computation will get affected.
>
>     1- Lars Lindemann, Lejun Jiang, Nikolai Matni, and George J. Pappas. Risk of stochastic systems for temporal logic specifications, 2022. URL https://arxiv.org/abs/2205.14523.
>
>     2- Alexandre Donzé and Oded Maler. Robust satisfaction of temporal logic over real-valued signals. In International Conference on Formal Modeling and Analysis of Timed Systems, pp. 92–106. Springer, 2010.
>
>     3- Pant, Yash Vardhan, Houssam Abbas, and Rahul Mangharam. "Smooth operator: Control using the smooth robustness of temporal logic." 2017 IEEE Conference on Control Technology and Applications (CCTA). IEEE, 2017.
>
>     4 - Nitish Srivastava, Geoffrey Hinton, Alex Krizhevsky, Ilya Sutskever, and Ruslan Salakhutdinov. Dropout: a simple way to prevent neural networks from overfitting. The journal of machine learning research, 15(1):1929–1958, 2014.

---

> ### Author Response · Authors · 2023-11-17
> **Response to Reviewer NFTL [Third part]**
>
> Weakness 7 :  Due to space restrictions, our description may hamper understanding the main scheme. The main idea is as follows:
>
> The $n^{th}$ time-step of the trajectory can be given by the following formula:
>
>     f( f( ... f( s(0), c(s(0)  ) , c(s(1) ), …), c(s(n)))
>
> In this trajectory, each “$c$” function is the neural network controller. The idea is to only consider the neural network parameters to be trainable parameters for selected time-points. This allows us to approximate the gradient of the robustness w.r.t. the same trainable parameters, but over fewer occurrences (in the gradient computation). We can view this as a form of drop-out [4](with the novel difference that we replace dropped out nodes/layers with an approximate value obtained from the parameters in the previous gradient computation step).
>
> We provide an illustrative example below to make the trajectory sampling more clear and then we answer two both questions you proposed:
>
> Let state, actions at time $k$ be $s(k), a(k)$ respectively, and feedback controller $a(k) = c (s(k))$ and the dynamics: $s(k+1) = f( s(k) , a(k))$.
>
> Let’s assume a trajectory of length $9$ over time-domain $T = \\{ i \mid 0 \le i \le 9\\}$, i.e.,
>
>     traj = { s(0), s(1), s(2), s(3), s(4), s(5), s(6), s(7), s(8), s(9) }
>
> where s(0) =1.15, and traj is generated with the set of actions:
>
>     act  ={a(0), a(1), a(2), a(3), a(4), a(5), a(6), a(7), a(8)}
>
> Suppose, we are at the iteration j=42 of backpropagation, and in this iteration, suppose T is sampled into M=3 subsets of N=3 time steps:
>
>     T1 = {0, 2, 4, 9},    T2 = {0, 5, 7, 8},   T3={0, 1, 3, 6}
>
> As you can see the union of T1, T2, T3 is T that means all the time steps are covered. The set T1, T2, T3 represent the sampled trajectories traj1, traj2, traj3 respectively where,
>
>     traj1 = { s(0), s(2), s(4), s(9)},     traj2 = { s(0), s(5), s(7), s(8)},     traj3 = { s(0), s(1), s(3), s(6)}.
>
> The interrelation between the states on a sampled trajectory is defined in a specific iteration of back-propagation algorithm.
>
> Suppose the NN parameters in iteration j=42 are  {1.2, 2.31, -0.92}. Using these control parameters and the knowledge that s(0)= 1.15, we can numerically evaluate the trajectory states (traj) and the sequence of control actions (act). Assume the sequence of actions at this iteration is computed as:
>
>     act = { 0, 0.1,  0.2,  0.3,  0.4,  0.5  0.6,  0.7,  0.8 }
>
> We then utilize these action values to define the sampled trajectories traj1, traj2, traj3.
>
> For traj1 we assume a(0) = c( s(0) ), a(2) = c( s(2) ), a(4) = c( s(4) ), are functions of states s(0), s(2), s(4) respectively, and we also assume the rest of control actions are fixed. i.e.
>
>     s(2)   = f(  f( s(0) , c( s(0) ) ) ,  0.1 )
>     s(4)   = f(  f( s(2) , c( s(2) ) ) ,  0.3 )
>     s(9)   = f(  f(  f(  f(  f( s(4) , c( s(4) ) ) ,  0.5 ) , 0.6 ) , 0.7 ) , 0.8 )
>
> For traj2 we assume a(0) = c( s(0) ), a(5) = c( s(5) ), a(7) = c( s(7) ) are  functions of states s(0), s(5), s(7) respectively, and we also assume the rest of control actions are fixed. i.e.
>
>     s(5)   = f(  f(  f(  f(  f( s(0) , c( s(0) ) ) ,  0.1 ) , 0.2 ) , 0.3 ) , 0.4 )
>     s(7)   = f(  f( s(5) , c( s(5) ) ) ,  0.6 )
>     s(9)   = f(  f( s(7) , c( s(7) ) ) ,  0.8 )
>
> For traj3 we assume a(0) = c( s(0) ), a(1) = c( s(1) ), a(3) = c( s(3) ) are  function of states s(0), s(1), s(3) respectively, and we also assume the rest of control actions are fixed. i.e.
>
>     s(1)   = f( s(0) , c( s(0) ) )
>     s(3)   = f(  f( s(1) , c( s(1) ) ) ,  0.2 )
>     s(6)   = f(  f(  f( s(3) , c( s(3) ) ) ,  0.4 ) , 0.5 )
>
> Now that we explained how we generate the sampled trajectories we answer to both of your questions in the following paragraph:
>
> As you can see from the example above, since the union of sets T1, T2 and T3 is T then, every single row of S will be approximated. In this specific example with scalar states, every single row in matrix S corresponds to a time step. Let S( k , :) denote the row corresponding to the $k^{th}$ time step. To approximate S(k, :), we follow a scheme defined in the example above. For instance, at time k=7 sampled in T2 ={ 0, 5 , 7 , 8}, (which defines traj2 = { s(0), s(5), s(7), s(8)}), S( 7 , : ) will be approximated from
>
>     s(5)   = f(  f(  f(  f(  f( s(0) , c( s(0) ) ) ,  0.1 ) , 0.2 ) , 0.3 ) , 0.4 )
>     s(7)   = f(  f( s(5) , c( s(5) ) ) ,  0.6 )
>     s(9)   = f(  f( s(7) , c( s(7) ) ) ,  0.8 )
>
> We know S( 7 , : ) is the derivative of s(7) with respect to the parameters of controller c . The accurate version of this derivative requires 8 multiplications of control parameters, since s(7) is the seventh state on the real trajectory, while the approximate derivative assumes s(7) is in traj2 and requires only 3 multiplications (as it is the second state in traj2). For longer trajectories, dramatically down-sampling the trajectory, thus helps address the exploding/vanishing gradients problem by effectively reducing the number of multiplications.

---

> > ### Comment · Reviewer_NFTL · 2023-11-22
> >
> > Thank you for the detailed response to my questions and comments. I do not see a revision here in open review - has one been posted?

---

> ### Author Response · Authors · 2023-11-22
> **Thanks for the reply**
>
> Hi,
>
> Thanks for the reply.
>
> Yes, the revision of submitted text document is ready for download. There is a PDF icon on the top of this webpage. Please click on that and download the updated text document.
>
> Regarding the responses to your valuable comments. We didn't add new revision from 3 days ago. Our final answers are ready for you to read in three parts, as you see above. Thanks for your time.

---

### Official Review · Reviewer_pDd8 · 2023-10-29

**Soundness:** 3 good
**Presentation:** 3 good
**Contribution:** 2 fair
**Rating:** 6
**Confidence:** 5

**Summary:**

This paper presents an algorithm of synthesizing neural policies for signal temporal logic specifications. Since RNNs over long
time horizon has problems of exploding and vanishing gradients, two main claimed features are proposed: smooth operators for robustness representation and a sampling-based approach to approximate the gradient.

**Strengths:**

1. The related work is well-written and the authors are aware of many recent developments.
2. The paper is well-presented and easy to follow.
3. The proposed features and algorithm are effective.
4. The benchmarking environments are standard and convincing in the community.

**Weaknesses:**

My main concern lies in the experimental comparison. I think the authors are recommended to compare with some other baselines because readers are not sure how good are the learned policies on benchmarks. I would suggest to compare with the standard MPC approach using Mixed Integer Linear Program (MILP) [1] to see if the learned policies are close to the optimal policies returned by MILP or not.

MILP has high computational complexity but still can be practically well-solved using tools like Gurobi --- so the benefits and potential drawbacks of neural net based STL synthesis over MILP should be clarified and also compared in practice.

[1] Vasumathi Raman, Alexandre Donzé, Mehdi Maasoumy, Richard M Murray, Alberto SangiovanniVincentelli, and Sanjit A Seshia. Model predictive control with signal temporal logic specifications. In Proc. of CDC, pp. 81–87. IEEE, 2014.

**Questions:**

I do not have question.

---

> ### Author Response · Authors · 2023-11-17
> **Reply to Reviewer PDd8**
>
> We thank for all of your valuable comments and suggestions. Here we answer to your comments titled as "Weakness" and "Question".
>
> Weakness 1 :   Comparing with greedy and bounded-horizon algorithms like MPC has several challenges. First, MPC-based methods like [1] scale very poorly with the time horizon, because in the MILP formulation, there is a new variable introduced for every time-step. For example, paraphrasing from [1]: For the robustness-based encoding of the MPC problem, $O(N \\cdot |P|)$ continuous variables are introduced (one per predicate per time step) to formulate the optimization problem, where $N$ is the number of time-steps, and $P$ is the set of signal predicates. For long time horizons, this number can be very large, and the scalability of MILP solvers greatly suffers.
>
> Weakness 2 :    We agree that MILP solvers have greatly improved over time, but they are still fundamentally limited because the problem they address is NP-hard. Furthermore, there is limited notion of an approximately optimal solution using MILP methods, whereas even when our gradient-based methods give sub-optimal answers (perhaps due to presence of local optima), as long as the STL2LB values are positive, we get a controller that guarantees that the system satisfies STL specifications over a chosen number of initial conditions over long time horizons.
>
>
> References:
>
> 1- Vasumathi Raman, Alexandre Donzé, Mehdi Maasoumy, Richard M Murray, Alberto SangiovanniVincentelli, and Sanjit A Seshia. Model predictive control with signal temporal logic specifications. In Proc. of CDC, pp. 81–87. IEEE, 2014.

---

### Official Review · Reviewer_Ga35 · 2023-10-31

**Soundness:** 2 fair
**Presentation:** 2 fair
**Contribution:** 2 fair
**Rating:** 5
**Confidence:** 4

**Summary:**

This paper addresses the problem of finding a controller satisfying a given Signal Temporal Logic (STL) specification in a given environment. The algorithm proceeds by sampling trajectories from a known environment given a parameterized policy after which a smoothened differentiable temporal logic structure is used to provide feedback. The policy is then updated using this feedback signal until a positive robustness score is reached indicating the specification is satisfied. The paper further introduces a technique to handle long horizon problems by approximating the gradient with fewer samples after identifying the critical predicate. Experimental results compare the effectiveness of these two methods over a range of task horizons and problems.

**Strengths:**

- Provides a lower bound to STL evaluation using differentiable computation graphs that fit well into neural network architectures.
- Addresses the vanishing gradient problem for long horizon tasks by means of sampling the gradients at given time steps.

**Weaknesses:**

- Lack of comparisons made to the state-of-the-art methods [1,2,3] for control using STL. The presented algorithms are shown without reporting results on enough competing methods. This brings into question the relative benefits of the given approach.
- The motivation for a smooth lower bound on the STL score is mentioned but not sufficiently justified (viz. empirically). It would be interesting to see how far the STL2NN method would work without the approximations provided by STL2LB in Algorithm 1. Another useful smoothing technique could be as introduced in [3].
- Assumes differentiability of the simulator environment and knowledge of its transition functions to calculate the policy parameter gradients. This may be infeasible in many problems.

References:

[1] Backpropagation through Signal Temporal Logic Specifications: Infusing Logical Structure into Gradient-Based Methods, Leung et al., WAFR 2020

[2]  Robust Counterexample-guided Optimization for Planning from Differentiable Temporal Logic, Dawson & Fan, 2022

[3] A Smooth Robustness Measure of Signal Temporal Logic for Symbolic Control, Gilpin et al., LCSS 2021

**Questions:**

1. In which steps in Algorithm 2 is the critical predicate $h^*$ and $k^*$ used? This is not entirely clear to me.
2. Is it possible to include comparisons to other algorithms in the same environment such as [1]? If not, why is that the case?
3. The introduction mentions  an RNN-based implementation, but this is not explained further in the text. Could there be a section in the appendix with more implementation details? Is there a clear benefit versus not using a fully connected network with the observation (and say the current time) as input being the policy?

---

> ### Author Response · Authors · 2023-11-17
> **Reply to Reviewer Ga35 [Part 1]**
>
> We thank for all of your valuable comments and suggestions. Here we answer to your comments titled as "Weakness" and "Question". And we start with "Weakness" section.
>
> Weakness:
>
> Weakness 1 :  All other papers using smooth semantics of STL do not use neural network based feedback controllers, and it is unclear how comparing against them can demonstrate the effectiveness of our approach. For example, in [3,7,1], the authors do not train a controller, but instead only obtain an open-loop control policy from a single initial state.  The only previous work that also trains closed-loop controllers is [5], and the summary of our comparison is in Section 4, under the heading, “Comparison.” Both our paper and [5] train controllers over a set of initial states.
>
> Reinforcement Learning based techniques such as [9, 10] are either tabular techniques that do not use NN controllers, or may not use smooth semantics [11],  may not be scalable on  long time horizons [5, 4, 12, 1 , 2, 3 ] (longest considered time horizon in these papers is 100 time steps).
>
> A separate study could be done where we compare only the function STL2LB with other notions of smooth semantics in the context of synthesizing open-loop control policies.  Superficially, the computational burden using STL2LB and other smooth semantics may look similar. However, during training, if robustness values of the signal predicates become high or low, then other smooth min/max versions give incorrect answers, potentially giving nonsensical values for the gradient. We added Appendix L to the updated text document that provides this comparison. Here we intentionally select a spacious environment to have large enough quantitative values for predicates which results in failure for [3],[4], and we also show that STL2LB successfully solves this problem.
>
>
> Weakness 2 :  Although we have already made a comparison between STL2LB and the smooth formula utilized in [5], we will also  include a comparison with the notion introduced in [3]. We were not aware of this paper at the time of submission, so we thank you for giving us a pointer to this very useful work.
>
> The value computed by STL2NN is the same as the robustness function in [6]. However, it is well known in the STL community that the semantics in [6] do not lead to an efficient training process due to the function not being differentiable (see the abstract of [4] and their experimental results).
>
>
> Weakness 3 :  We agree that the assumption to access the transition function may be infeasible for many problems. However, we can combine our technique with methods for learning dynamical system models which are common in control theory and model based reinforcement learning.
>
>
> Questions:
>
> Question 1 :  The critical time is used in line 11 of Alg.2. Once the critical predicate h* is found and the critical time k* is located on the trajectory, we approximate the gradient of h* with respect to control parameters. To that end, we sample arbitrary time steps between k=0 and k=k* while we make it certain that k=0 and k=k* are both included in the set of sampled times. We then generate the sampled trajectory, and finally we utilize this sampled trajectory to compute the approximate gradient of h* with respect to the control parameters. We note that the use of critical predicates and critical times allows us to approximate the gradient. This idea of sampled trajectory and gradient sampling is similar in spirit to the idea of drop-out layers [8].
>
>
> Question 2 :  The work in [1] uses smooth semantics of STL but does not use neural network based feedback controllers, but instead, it focuses on obtaining an open-loop control policy from a single initial state.It is unclear how comparing against this work can demonstrate the effectiveness of our approach. Furthermore, there is no explicit consideration of long horizon temporal tasks in [1], and it is unlikely that the technique in [1] will cope with numerical issues introduced by long horizon specifications. We will include new results that experimentally demonstrate this and update the rebuttal with a summary.
>
>
> Question 3 :
>
> We agree that we could have explained this further.  Our technique does not use an RNN-based controller. What we want to highlight is that the composed system, consisting of the plant dynamics, the neural network-based controller unrolled over the task horizon creates a structure that is very similar to that of sequence-to-sequence RNNs. Thus, the challenges of training such RNN structures are also challenges that we face. We will make this clearer in the text to avoid confusion.
>
> We added more details in the appendix K of the updated text document, with an illustration that shows the structure of the control system that we consider. In fact, both your suggestions already exist in the paper: the control action is a fully connected feedforward neural network and is only dependent on the state at a given time, and the value of the time step itself.

---

> ### Author Response · Authors · 2023-11-17
> **Reply to Reviewer Ga35 [Part 2] (References)**
>
> 1- Karen Leung, Nikos Arechiga, and Marco Pavone. Back-propagation through signal temporal logic specifications: Infusing logical structure into gradient-based methods. In Steven M. LaValle, Ming Lin, Timo Ojala, Dylan Shell, and Jingjin Yu (eds.), Algorithmic Foundations of Robotics XIV, pp. 432–449. Springer, 2021.
>
>      2- Robust Counterexample-guided Optimization for Planning from Differentiable Temporal Logic, Dawson & Fan, 2022
>
>      3- A Smooth Robustness Measure of Signal Temporal Logic for Symbolic Control, Gilpin et al., LCSS 2021
>
>      4- Pant, Yash Vardhan, Houssam Abbas, and Rahul Mangharam. "Smooth operator: Control using the smooth robustness of temporal logic." 2017 IEEE Conference on Control Technology and Applications (CCTA). IEEE, 2017.
>
>      5- Yaghoubi, Shakiba, and Georgios Fainekos. "Worst-case satisfaction of stl specifications using feedforward neural network controllers: a lagrange multipliers approach." ACM Transactions on Embedded Computing Systems (TECS) 18.5s (2019): 1-20.
>
>      6- Alexandre Donzé and Oded Maler. Robust satisfaction of temporal logic over real-valued signals. In International Conference on Formal Modeling and Analysis of Timed Systems, pp. 92–106. Springer, 2010.
>
>      7- Karen Leung, Nikos Aréchiga, and Marco Pavone. Backpropagation for parametric stl. In 2019 IEEE Intelligent Vehicles Symposium (IV), pp. 185–192. IEEE, 2019.
>
>      8- Nitish Srivastava, Geoffrey Hinton, Alex Krizhevsky, Ilya Sutskever, and Ruslan Salakhutdinov. Dropout: a simple way to prevent neural networks from overfitting. The journal of machine learning research, 15(1):1929–1958, 2014.
>
>      9- Aksaray, Derya, et al. "Q-learning for robust satisfaction of signal temporal logic specifications." 2016 IEEE 55th Conference on Decision and Control (CDC). IEEE, 2016.
>
>      10- Li, Xiao, Cristian-Ioan Vasile, and Calin Belta. "Reinforcement learning with temporal logic rewards." 2017 IEEE/RSJ International Conference on Intelligent Robots and Systems (IROS). IEEE, 2017.
>
>      11- Balakrishnan, Anand, and Jyotirmoy V. Deshmukh. "Structured reward shaping using signal temporal logic specifications." 2019 IEEE/RSJ International Conference on Intelligent Robots and Systems (IROS). IEEE, 2019.
>
>       12- Karen Leung, Nikos Aréchiga, and Marco Pavone. Backpropagation for parametric stl. In 2019 IEEE Intelligent Vehicles Symposium (IV), pp. 185–192. IEEE, 2019.

---

> > ### Comment · Reviewer_Ga35 · 2023-11-21
> > **Thanks for the response**
> >
> > Thanks to the authors for the clarification on the critical point and RNN controller. I also appreciate the additional experiments in Appendix L regarding other smoothness measures.
> >
> > How were these comparison experiments (App. L) tuned? Are they the best over a range of hyperparameters such as learning rate? Are the curves in Fig 14 averaged over multiple random seeds? Such information would give these results greater merit.
> >
> > Based on the points raised by the other reviewers I am inclined to keep my evaluation.

---

> > > ### Author Response · Authors · 2023-11-21
> > > **Thanks for the reply**
> > >
> > > Thanks for your reply.
> > >
> > > Your first question is :
> > >
> > >         How were these comparison experiments (App. L) tuned? Are they the best over a range of hyperparameters such as learning rate?
> > >
> > > Answer:
> > >
> > > We know that, for small values of $b$, the smooth robustness can not estimate the real robustness well, that is a drawback. On the other hand, for large values of $b$ the smooth robustness will generate non-smooth behaviors as it becomes identical to non-smooth robustness. Therefore, tuning for $b$ will be done through a trade-off.
> > >
> > > Regarding tuning of learning rate, we need to make it clear that we use the function adamupdate() in MATLAB. This function automatically takes care of learning rate and momentum coefficients and the other training parameters.
> > >
> > > For this experiment, we utilized our toolbox for Algorithm 1 and trained the controllers in 3 different scenarios. In all the scenarios, we started the training with a unique guess for initial control parameters.
> > >
> > > In the first scenario, we utilized Algorithm 1, and we hard coded the smooth semantics proposed by [3] for gradient computation. Regarding the size of the quantitative semantics for predicates, the smooth semantics in [3] resulted in NaN for gradient when $b \\geq 8$. Therefore, for tuning the parameter $b$ we ranged it from $1$ to $7$ and reported the learning curves for each case on figure 14. Among these experiments, the case $b=4$ resulted in best answer but was not satisfying the STL spec.
> > >
> > > In the second scenario, we utilized Algorithm 1, and we hard coded the smooth semantics proposed by [4] for gradient computation. Our experimental results show, similar to [3], this smooth semantics is unable to handle $b \geq 8$, thus we ranged the parameter $b$ from $1$ to $7$ and reported the learning curves on figure 14. Again, $b=4$ resulted in the best answer but was not satisfying the STL spec.
> > >
> > > In the third scenario, we picked up $b=4$ and generated STL2LB and utilized it in Algorithm 1 for training. Although the previous techniques did not result in positive robustness, this smooth function provides us positive robustness within 2700 gradient steps.
> > >
> > > The point of this experiment was to show the computational problem in [3],[4] and to report how can this computational problem be problematic in a training process. As you see, for $b=8$ both [3], [4]  result in NaN for gradient due to computational error. This implies although for b=1,2,3,4,5,6,7 the gradient is not NaN, but the computational problems still exist which results in non-informative gradients.
> > >
> > > However, STL2LB have no computational issue and can handle every possible hyperparameter $b$. In addition to $b=4$, we also trained the controller with STL2LB for $b=15$, and it provided us positive robustness within $3300$ gradient steps.
> > >
> > > Your second question is:
> > >
> > >      Are the curves in Fig 14 averaged over multiple random seeds?
> > >
> > > Answer:
> > >
> > > The curves in Fig 14 are an average over the five sampled initial states that we utilized in the training process. These initial states are:
> > >
> > >     s0 = [0 ;  0],   s0 = [2 ; 0],  s0 = [0 ; 2],  s0 = [ 2 ; 2],  s0 = [ 1 ; 1]
> > >
> > >
> > > Your suggestion:
> > >
> > >     Such information would give these results greater merit.
> > >
> > > Thanks for your suggestion, we will include the provided information in appendix L to make this comparison more clear.

---

### Official Review · Reviewer_xgYh · 2023-11-05

**Soundness:** 2 fair
**Presentation:** 1 poor
**Contribution:** 2 fair
**Rating:** 3
**Confidence:** 2

**Summary:**

The authors propose an algorithm for optimizing neural network control policies to satisfy STL specifications on their behavior. They build on prior work in differentiating through a smoothed approximation of the STL robustness signal, adding two core contributions. First, they propose a smooth approximation scheme which guarantees a lower bound on the true robustness while avoiding numerical issues. Second they propose a gradient approximation scheme to manage issues with gradients exploding or vanishing over long temporal horizons, involving evaluating the gradient of the trajectory wrt to policy parameters only at certain timesteps.

**Strengths:**

- The paper tackles an important challenge of enabling STL-based training over long temporal horizons without running into the challenges such as vanishing gradients.
- The background on STL and STL robustness was thorough and helpful, though this may have come at the expense of having not enough room to clearly explain the core contributions of this work.
- The experimental results on show promising results in effectively satisfying STL formulae for reasonably high-dimensional systems over long temporal horizons.

**Weaknesses:**

- The paper was difficult to follow. In particular, section 3.3 detailing the sampling based approximation of the gradient was quite hard to understand. A figure to help illustrate the trajectory subsampling approach would significantly improve the clarity of the paper.
- The impact of STL2LB as compared to other strategies to smooth STL formulae was not clearly demonstrated.
- The experimental comparisons were very limited: quantitative comparisons were only presented against ablations of the proposed approach, and not against many of the other cited works on training NN policies to satisfy STL objectives. In addition, there was no comparison against an approach which used the critical predicate-based time sampling, but without the waypoint functions. Thus it was not clear what improvement the critical predicate-based sampling strategy had over random time sampling.
- The paper would be strengthened with theoretical results detailing what factors impact the quality of the sampling-based gradient approximation.

**Questions:**

- Do you have empirical results investigating how well the sampling-based gradient strategy approximates the true gradient?
- The solution to the computational problems with the swish and softplus functions seems to break differentiability, especially in the case of the swish function. Would this lead to issues during optimization?

---

> ### Author Response · Authors · 2023-11-15
> **Reply to Reviewer xgYh [First Part]**
>
> We thank all reviewers for their valuable comments and suggestions. Here we respond to the points you provided us for weakness points and questions.
>
> Weakness:
>
> Weakness 1 :  The gradient approximation proposed in this paper can be viewed as a form of drop-out [1]. In drop-out, while neurons are used for back-propagation with some probability, in our setting, we replace entire layers with an approximate value ( A simple numerical example is also provided for  Reviewer 4 ). We included a figure (See figure 2) in Appendix A  for an example of gradient approximation technique to add clarity to this discussion.
>
> Let me explain our gradient approximation method from the perspective of drop-out layers [1]. Considering NN controllers rolled out over the trajectory, in drop-out layers, we remove randomly selected nodes from a neural network (in our case, this corresponds to removing random nodes from arbitrary time-points of the NN representing the controller rolled out over the trajectory horizon). Drop-out requires the node to be absent in both the forward pass and the backward pass in the backpropagation algorithm. To alleviate the problem of vanishing and exploding gradients, instead of picking arbitrary neurons in the controller (at arbitrary time-points), we sample certain time-steps and select all of the controller neurons in that time-step.
> Due to the reduced number of multiplications, this alleviates this problem of vanishing/ exploding gradients. However, we cannot simply remove all controller nodes at a specific time step, as the resulting composed NN can become disconnected. Thus, when we drop out selected neurons, we replace those neurons with a node representing a constant function whose value is equal to the evaluation of the controller unit in the forward pass. This argument is the main motivation behind the definition of sampled trajectory.
>
>
> Weakness 2 :  All other papers using smooth semantics of STL do not use neural network based feedback controllers, and it is unclear how comparing against them can demonstrate the effectiveness of our approach. For example, in [3,4,5,8], the authors do not train a controller, but instead only obtain an open-loop control policy from a single initial state.  The only previous work that also trains closed-loop controllers is [2], and the summary of our comparison is in Section 4, under the heading, “Comparison.” Both our paper and [2] train controllers over a set of initial states. Reinforcement Learning based techniques such as [9, 10] are either tabular techniques that do not use NN controllers, or may not use smooth semantics [11], and may  not scale to long time horizons [2, 3, 4, 5 , 6, 7, 8 ] (longest considered time horizon in these papers is 100 time steps).
>
> A separate study could be done where we compare only the function STL2LB with other notions of smooth semantics in the context of synthesizing open-loop control policies.  We expect that the computational efficiency to be similar as other approaches; however, we remark that only our approach will guarantee soundness, i.e., if given a single initial state a control policy is obtained that has positive value of the STL2LB function, then the system behavior under this policy is guaranteed to satisfy the STL formula. This is not true for other smooth semantics. We argue that this study is somewhat orthogonal to the objectives of our paper (scaling neural network controllers to satisfy long-horizon tasks, and pushing the training algorithm towards satisfying safety guarantees).
>
> Weakness 3 :  We omitted the ablation study that involved using only critical predicate-based time sampling but not waypoint functions due to space restrictions. Our study shows that critical predicate-based time sampling still outperforms random time-sampling (even without waypoint functions), and adding waypoints functions boosts the performance even further. We have included new experiments to demonstrate this aspect. Please see Appendix G in the updated text document.
>
>
> Weakness 4 :  This is indeed a very interesting suggestion that we will consider for future work. Critical predicate-based sampling is quite property and system dependent, it is unclear if we can make general statements about the training algorithm for all STL formulas and environments. However, the general idea of dropping layers in an unrolled recurrent structure can be explored theoretically. One idea is to characterize the worst-case error introduced by the drop-out-like approximations [1] (that we introduce), and show that our training algorithm converges to the same (local) optimum as a training algorithm that does not use any dropout. We will look at the literature from stochastic gradient optimization for future work.

---

> ### Author Response · Authors · 2023-11-15
> **Reply to  Reviewer xgYh  [Second part]**
>
> Questions
>
> Question 1 :     We remark that the true gradient involves backpropagation over the entire trajectory, which often leads to vanishing or exploding gradients in long horizon specifications. Our sampling-based gradient, on the other hand, does not have this problem. By definition, these two gradients are not expected to be close to each other in value; however, we expect their directions to be approximately similar. However, it is still difficult to check if the inner product of the unit vectors in each gradient’s direction is 1 (due to the exploding/vanishing gradient problem). Hence, our approach to show this is through the ablation study in Table 1. We can argue that drop-out based techniques [1] have efficacy due to similar reasons.
>
> Question 2:  You are right, the modified swish function is not differentiable. However, in practice, we observe that we can pick the hyperparameter $\tau$ to guarantee that the derivatives are numerically close when we approach the point of non-differentiability from either side. Recall that
> \\begin{equation}
> \\text{if $\\zeta > -\\tau/b$}  \\quad \\text{then} \\quad \\widetilde{swish}(\\zeta)=
> swish(\\zeta),
> \\end{equation}
> \\begin{equation}
> \\text{if $\\zeta < -\\tau/b$}  \\quad  \\text{then} \\quad \\widetilde{swish}(\\zeta)=0
> \\end{equation}
> For example, we can set $ \\tau = 720$. For $b=10$, then the point of non-differentiability is when x = -72. At this point, derivative of $\\widetilde{swish}(x)$  is  $-1.46\times10^{-310}$ (approaching from the *right*) and 0 approach from the left, which is numerically an insignificant difference, and smaller than the least number that can be represented in double precision in MATLAB. Similarly, for modified softplus, the derivatives from either side at x = 72 are identical from the perspective of numerical precision. However in our experiments we set $ \\tau$ =50, and for b=10 the point of non-differentiability is when $x=-5$, At this point the derivative of $ \\widetilde{swish}(x)$ is  $-9.4509 \\times 10^{-21}$, while the derivative of $y=0$ at $x=-5$ is equal to 0. Thus, we  conclude a difference of $9.4509 \\times 10^{-21}$, in practice, may not be problematic regarding differentiability. In addition, a simple calculation for the derivative of $\\text{softplus}(x )$ at $x=5$, $b=10$ implies the slope of $y=softplus(x)$ at $x=5 $ is $(1 - 1.9287 \\times 10^{-22})$ which is very close to $1$ this is while the slope of $y=x$ is exactly equal to $1$. Thus, we conclude a difference of $1.9287 \\times 10^{-22}$, in practice, may not provide computational error related to differentiability.
>
>
> References:
>
>     1 - Nitish Srivastava, Geoffrey Hinton, Alex Krizhevsky, Ilya Sutskever, and Ruslan Salakhutdinov. Dropout: a simple way to prevent neural networks from overfitting. The journal of machine learning research, 15(1):1929–1958, 2014.
>
>     2- Yaghoubi, Shakiba, and Georgios Fainekos. "Worst-case satisfaction of stl specifications using feedforward neural network controllers: a lagrange multipliers approach." ACM Transactions on Embedded Computing Systems (TECS) 18.5s (2019): 1-20.
>
>     3- Pant, Yash Vardhan, Houssam Abbas, and Rahul Mangharam. "Smooth operator: Control using the smooth robustness of temporal logic." 2017 IEEE Conference on Control Technology and Applications (CCTA). IEEE, 2017.
>
>     4- Karen Leung, Nikos Aréchiga, and Marco Pavone. Backpropagation for parametric stl. In 2019 IEEE Intelligent Vehicles Symposium (IV), pp. 185–192. IEEE, 2019.
>
>     5- Karen Leung, Nikos Arechiga, and Marco Pavone. Back-propagation through signal temporal logic specifications: Infusing logical structure into gradient-based methods. In Steven M. LaValle, Ming Lin, Timo Ojala, Dylan Shell, and Jingjin Yu (eds.), Algorithmic Foundations of Robotics XIV, pp. 432–449. Springer, 2021.
>
>     6- Robust Counterexample-guided Optimization for Planning from Differentiable Temporal Logic, Dawson & Fan, 2022
>
>     7- A Smooth Robustness Measure of Signal Temporal Logic for Symbolic Control, Gilpin et al., LCSS 2021
>
>     8- Karen Leung, Nikos Arechiga, and Marco Pavone. Back-propagation through signal temporal logic specifications: Infusing logical structure into gradient-based methods. In Steven M. LaValle, Ming Lin, Timo Ojala, Dylan Shell, and Jingjin Yu (eds.), Algorithmic Foundations of Robotics XIV, pp. 432–449. Springer, 2021.
>
>     9- Aksaray, Derya, et al. "Q-learning for robust satisfaction of signal temporal logic specifications." 2016 IEEE 55th Conference on Decision and Control (CDC). IEEE, 2016.
>
>     10- Li, Xiao, Cristian-Ioan Vasile, and Calin Belta. "Reinforcement learning with temporal logic rewards." 2017 IEEE/RSJ International Conference on Intelligent Robots and Systems (IROS). IEEE, 2017.
>
>     11- Balakrishnan, Anand, and Jyotirmoy V. Deshmukh. "Structured reward shaping using signal temporal logic specifications." 2019 IEEE/RSJ International Conference on Intelligent Robots and Systems (IROS). IEEE, 2019.

---

### Meta-Review · Area_Chair_DVGD · 2023-12-14

**Metareview:**

The paper introduces a model-based approach for safe control. Training policies for autonomous agents to satisfy specific task objectives and safety constraints expressed in Signal Temporal Logic. The strength of the paper is gradient approximation algorithm based on gradient sampling, which improves the quality of the stochastic gradient and enables scalable back-propagation over long time horizon trajectories.

There is a need for further validation and comparison with existing methods. Experiments on a wider range of benchmark problems are required to establish its superiority over other approaches. As an experimental work, the practical implementation and computational efficiency in real-world scenarios should be included.

**Justification For Why Not Higher Score:**

There is a clear lack of comparison with existing methods.

**Justification For Why Not Lower Score:**

N/A

---

### Decision · Program_Chairs · 2024-01-16

Reject